# MAFE: Enabling Equitable Algorithm Design in Multi-Agent Multi-Stage Decision-Making Systems

## Abstract

Algorithmic fairness is often studied in static or single-agent settings, yet many real-world decision-making systems involve multiple interacting entities whose multi-stage actions jointly influence long-term outcomes. Existing fairness methods applied at isolated decision points frequently fail to mitigate disparities that accumulate over time. Although recent work has modeled fairness as a sequential decision-making problem, it typically assumes centralized agents or simplified dynamics, limiting its applicability to complex social systems. We introduce **MAFE**, a suite of *Multi-Agent Fair Environments* designed to simulate realistic, modular, and dynamic systems in which fairness emerges from the interplay of multiple agents. We demonstrate MAFEs in three domains—loan processing, healthcare, and higher education—supporting heterogeneous agents, configurable interventions, and fairness metrics. The environments are open-source and compatible with standard multi-agent reinforcement learning (MARL) libraries, enabling reproducible evaluation of fairness-aware policies. Through extensive experiments on cooperative use cases, we demonstrate how MAFE facilitates the design of equitable multi-agent algorithms and reveals critical trade-offs between fairness, performance, and coordination. MAFE provides a foundation for systematic progress in dynamic, multi-agent fairness research.

## 1 Introduction

As machine learning (ML) systems increasingly shape decisions in critical domains, such as lending, healthcare, and education, concerns have intensified about their potential to exacerbate social inequities Sweeney (2013); Angwin et al. (2016); Larson et al. (2016); Buolamwini & Gebru (2018). The field of *algorithmic fairness* seeks to design interventions that not only mitigate bias at the point of decision but also prevent disparities from compounding over time.

While early approaches focused on static definitions of fairness—targeting group-level Kamiran & Calders (2012); Hardt et al. (2016), individual-level Dwork et al. (2012), and causal Kusner et al. (2017); Coston et al. (2020) biases—these criteria often fall short in dynamic settings. For example, a healthcare system that ensures equal treatment at diagnosis may still produce inequitable long-term outcomes if certain populations face barriers to follow-up care Liu et al. (2018); D'Amour et al. (2020). Addressing such evolving disparities demands frameworks that capture sequential decisions and their cumulative effects.

Recent works model fairness through sequential lenses, using Markov Decision Processes (MDPs) Yin et al. (2024); Xu et al. (2024) or structural causal models Hu & Zhang (2022). However, these methods generally assume a single decision-maker operating in isolation. In contrast, real-world systems are multi-actor: insurers, hospitals, and government agencies jointly influence population health; schools, employers, and regulators together shape educational equity. Capturing such systems requires a shift to multi-agent formulations where fairness is not a property of one decision, but of distributed interactions across agents.

**Yet progress in this direction is constrained by the absence of realistic, modular environments for evaluating fairness in multi-agent systems.** Existing platforms typ-

ically assume centralized control, lack support for heterogeneity across agents, or oversimplify social dynamics, limiting their utility for fairness-aware algorithm design.

To bridge this gap, we introduce **MAFE** [1]—a benchmark suite of *Multi-Agent Fair Environments* for modeling dynamic decision-making systems where fairness arises from the interactions of multiple agents. Each MAFE is a modular and extensible simulation of a social system, featuring heterogeneous agents, configurable disparities, and evolving population dynamics. Designed to support algorithm development and empirical evaluation, MAFE offers a principled testbed for fairness-aware multi-agent learning.

**Summary of Contributions.** By introducing **MAFE** as a benchmark suite of *Multi-Agent Fair Environments* for evaluating fairness-aware policies in dynamic, multi-agent settings, we provide the following key contributions:

- **Framework and Benchmarks.** We propose the MAFE framework and instantiate three open-source environments—`MAFE-Loan`, `MAFE-Health`, and `MAFE-Edu`—that model equity challenges across social domains.

- **Fairness-Aware Modeling and Evaluation.** We define a cooperative use case and formalize fairness-aware optimization objectives that capture long-term system equity. We introduce diagnostic metrics and illustrate how MARL algorithms can be adapted to equity-driven reward structures through a representative implementation.

- **Empirical Validation.** We evaluate the behavior of a representative MARL algorithm in MAFE environments, offering reproducible baselines and highlighting trade-offs between fairness and utility.

## 2 Related Works

### 2.1 Single-Agent Long-Term Fairness.

To overcome the limitations of static fairness formulations, several approaches have re-framed fairness as a dynamic systems problem. Effort-based fairness analyzes the differing efforts required by groups to achieve outcomes Heidari et al. (2019); Guldogan et al. (2022), while causal models use structural causal models and interventions to introduce fairness Hu et al. (2020); Hu & Zhang (2022). Another approach incorporates fairness within dynamic systems through reinforcement learning (RL), with early work using multi-armed bandits Joseph et al. (2016) and recent efforts employing Markov Decision Processes (MDPs). Puranik et al. (2024) introduce the Fair-Greedy policy in an admissions case study, balancing applicants' scores with group proportions. Yin et al. (2024) frame the long-term fairness RL problem to maximize profits while minimizing unfairness, measured by regret and distortion. To address temporal bias, Xu et al. (2024) propose a fairness measure based on the ratio-after-aggregation and modify the proximal policy optimization algorithm (PPO) to satisfy this constraint. Though these works reduce temporal disparities, they do not analyze their source. Deng et al. (2024) use causal analysis to trace sources of inequality over time. While these works extend static fairness to long-term outcomes, Hu et al. (2023) argue that long-term fairness should focus on the convergence of input feature distributions, proposing a PPO variant with pre-processing and regularization to balance short- and long-term fairness.

### 2.2 Multi-Agent Long-Term Fairness.

In systems with multiple decision-making entities, modeling fairness explicitly across agents becomes crucial for understanding their interventions and their effects on system dynamics. Several studies have explored fairness in multi-agent contexts. Jiang & Lu (2019) introduce the Fair-Efficient Network, a hierarchical RL model where homogeneous agents aim to balance fairness and efficiency. Zheng et al. (2022) use two-level deep RL to design agents that reduce income inequality via taxation and redistribution, with equity measured by the

---

[1] We release our anonymized codebase at `https://anonymous.4open.science/r/MAFE_Environments-88CA/README.md`

Gini Index. Reuel & Ma (2024) provide a survey on fairness in RL, covering both single- and multi-agent systems. They highlight key gaps, such as fairness in RL from human feedback, and emphasize the challenges of ensuring fairness in dynamic real-world environments, which underscores the need for realistic simulation environments.

### 2.3 Long-Term Fairness Environments.

A major challenge in long-term fairness research is designing appropriate environments for measuring, simulating, and assessing fairness algorithms. Among the growing body of research on long-term fairness, some works have introduced environments that consider the complexities of real-world decision-making. For example, D'Amour et al. (2020) introduce lending and attention environments, while Atwood et al. (2019) focus on infectious disease environments. However, these environments are single-agent based. Real-world systems, by contrast, often consist of multiple interacting entities that influence outcomes. By not explicitly modeling these entities as agents, such environments limit the ability to flexibly analyze the various forms of intervention and the effects that these different entities may have on the system's underlying dynamics.

Although there are existing multi-agent fair environments focusing on taxation and economic policy Jiang & Lu (2019); Zimmer et al.

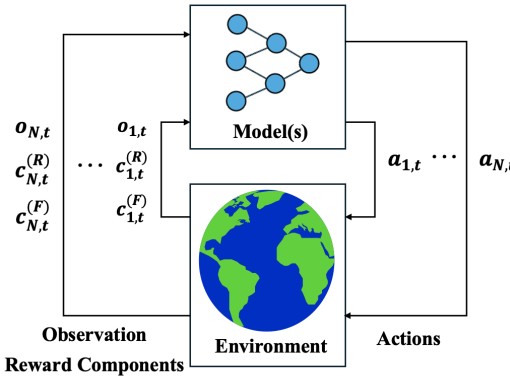

Figure 1: **MAFE Diagram.** Model(s) produce actions that are imported to the environment and taken by agents. This leads to state transition within the environment that produces a set collection of observations, rewards, and fairness components for each agent which are output by the environment for the model(s) to use to produce actions in the next time step.

(2021); Grupen et al. (2022), they typically assume homogeneous agents Wong et al. (2023); Aloor et al. (2024), create abstract toy environments Jiang & Lu (2019); Zimmer et al. (2021) not based on real data, or emphasize theoretical analysis behind fairness algorithms Ju et al. (2023). While such agents can, in principle, learn to optimize for group-level fairness (e.g., worst-case outcomes over labeled subgroups), these environments typically lack the population-level structure needed to model disparities across social groups. Fairness is often framed at the agent level, limiting their ability to capture long-term group-level dynamics and feedback effects. Additionally, their environments are simpler compared to real-world social systems, where stakeholders in fields such as healthcare and finance have diverse decision-making processes. Simply retrofitting synthetic group labels into these environments fails to capture the distributional dynamics and systemic disparities that fairness research aims to address. In contrast, our proposed framework supports **heterogeneous agents** targeting fairness across the populations served by these agents, an essential distinction in domains like healthcare, lending, and education, where equity concerns revolve around real-world outcomes for individuals with socially salient attributes , ensuring equitable outcomes across demographic groups. Furthermore, our MAFEs are built from **real-world datasets** and explicitly model multi-agent pipelines with demographically structured populations. While Zheng et al. (2022) environment offers a detailed model, its context is restricted to economic outcomes. Our framework spans multiple domains—including finance, healthcare, and education—each requiring tailored approaches and supporting multiple fairness measures across diverse contexts.

## 3 Formalizing Fairness in Multi-Agent Systems

**Motivation.** Many real-world fairness challenges—such as disparities in healthcare, education, or access to credit—are shaped by the sequential, interdependent actions of multiple decision-makers. These scenarios are naturally modeled as decentralized systems, where multiple agents, each with partial observability and localized goals, interact in a shared environment. While decentralized partially observable Markov decision processes (Dec-

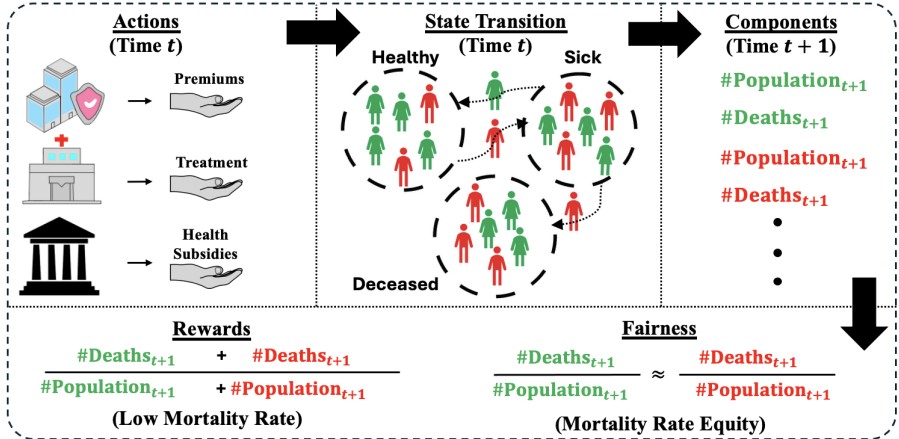

Figure 2: **Illustration of** `MAFE-Health`. An example illustrating how agents' actions in `MAFE-Health` affect the underlying health states in the population. By tracking indicators provided by the component functions, we can construct reward and fairness measures.

POMDPs) offer a suitable formalism for such settings, they lack explicit mechanisms for flexibly modeling fairness objectives and assessing social disparities.

**The MAFE Framework.** To address this, we propose the **Multi-Agent Fair Environment (MAFE)** framework—an extension of Dec-POMDPs that integrates fairness-aware reward design and diagnostic metrics for social disparity. A MAFE is defined by the tuple $\langle \mathcal{N}, \mathcal{S}, \{\mathcal{A}_n\}, \{\mathcal{O}_n\}, T, \gamma, \{c_n^{(R)}\}, \{c_n^{(F)}\}\rangle$, where $\mathcal{N}$ denotes the set of $N$ agents, $\mathcal{S}$ the global state space, $\mathcal{A}_n$ and $\mathcal{O}_n$ the action and observation spaces for agent $n$, $T$ the transition function over joint actions, and $\gamma$ the discount factor. $c_n^{(R)}$ and $c_n^{(F)}$ respectively denote the reward and fairness component functions for agent $n$. Figure 1 provides a diagram that illustrates the MAFE framework.

Unlike standard reward functions, which output a single scalar value per timestep, our component functions produce structured vectors of interpretable scalar quantities, such as counts or totals (e.g., number of deaths, population size). These are what we refer to as decomposable primitives—raw elements from which composite metrics like rates or disparities can later be constructed. This distinction is crucial: exposing these primitives allows for flexible and customizable evaluation. For instance, one can compute either a global mortality rate over time (total deaths divided by total population) or the average of per-time-step mortality rates, because both inputs (deaths and population) are available separately at each step. In contrast, standard Dec-POMDPs that incorporate fairness directly into rewards typically produce pre-aggregated composite values, which support only the latter, since such outputs cannot be decomposed into their underlying components. Without access to these base elements, defining temporally aggregated or alternative fairness metrics becomes difficult or even impossible within such models. This is the core motivation behind our component function formulation: to provide the flexibility needed for richer and more expressive fairness evaluations.

**Illustrative Example: Healthcare MAFE.** Consider a healthcare setting illustrated in Figure 2, which contains three agents: an insurance provider (agent 1), a hospital (agent 2), and a central planner (agent 3). The insurance provider sets premiums, the hospital allocates beds to a sick population, and the central planner manages public investment. These agents operate with different observations and control levers, yet their combined actions affect population health outcomes.

To illustrate a reward component function, consider the hospital, whose primary objective is to reduce mortalities. Instead of outputting a scalar mortality rate, the hospital's reward component function, $c_2^{(R)}$, might output a vector of the number of deaths and the total

population at each time step:

$$\left[\#\text{Deaths}_t, \quad \#\text{Population}_t\right]^T.$$

These decomposable values allow the hospital to compute mortality rates and track performance over time, providing more flexibility than pre-aggregated metrics.

For a fairness component function, consider the central planner, who aims not only to improve overall health outcomes but also to ensure equity across geographic regions. Suppose there are two such regions—A and B. The planner's fairness component function, $c_3^{(F)}$, might output the number of deaths and the population size in each region at time $t$:

$$\left[\#\text{Deaths}_t^A, \#\text{Deaths}_t^B, \#\text{Population}_t^A, \#\text{Population}_t^B\right]^T.$$

With access to these raw counts, the planner can compute region-specific mortality rates and monitor disparities, enabling more targeted and equitable public health investments.

**Modeling Flexibility.** A central strength of the MAFE framework is its flexibility in capturing complex multi-agent dynamics. Reward and fairness component functions can be tailored to reflect diverse agent roles, information structures, and objectives. MAFE supports both cooperative and non-cooperative settings, allowing agents to share goals or pursue distinct—and potentially conflicting—objectives. For instance, in a healthcare domain, an insurance provider might aim to minimize costs, a hospital might focus on patient recovery, and a central planner might prioritize equity across regions. Each agent can be assigned its own reward and fairness component functions, enabling fine-grained definitions of success that reflect their roles. MAFE also accommodates heterogeneous observation and action spaces: the insurer may observe population-wide data, while the hospital sees only admitted patients. Finally, by exposing decomposable base elements, MAFE enables both step-wise and temporally aggregated evaluation metrics—such as overall recovery rates or disparities across demographic groups—supporting a wide range of fairness analyses. For experimental clarity, we focus on cooperative MAFEs, leaving extensions to richer strategic settings for future work.

## 4 Instantiating MAFE in Social Domains

**Domain Coverage.** We construct three domain-specific environments using the MAFE framework: `MAFE-Health`, `MAFE-Loan`, and `MAFE-Edu`. Each models a real-world social system involving multiple stakeholders whose coordinated actions shape long-term equity. To ensure realism, our environment instantiations leverage publicly available datasets and domain-specific models, as detailed in Appendices F-H.

**Realism of MAFEs.** MAFEs emphasize realism in three different ways. First, MAFEs leverage Lending Club, IPUMS, NCES, and CDC datasets to provide raw attributes for individual feature vectors (loan applicants, patients, students). Second, populations are initialized by sampling from real feature distributions, ensuring agents train on realistic demographic and economic patterns. Third, relationships between features and key outcome indicators are derived via regression on real data, then resampled and slightly amplified to create controlled but realistic structural disparities across groups, as detailed in Appendix D.1. We now provide an overview of each environment.

***MAFE-Health.*** This environment simulates a population with evolving health states and three decision-making agents:

- **Insurance Agent**: Offers insurance coverage at a cost, influencing individuals' access to care.
- **Hospital Agent**: Allocates hospital beds to sick individuals based on demand and capacity.
- **Central Planner**: Invests in hospital infrastructure, public health programs, and insurance subsidies.

Individuals transition between health states (e.g., healthy, sick, dead) based on agent decisions and environmental dynamics. Geographic disparities in outcomes may arise from localized policies and resource constraints.

***MAFE-Loan.*** This environment simulates a financial system where individuals apply for, receive, and repay loans under the influence of three decision-making agents:

- **Admissions Agent**: Approves or rejects loan applications based on applicant profiles.
- **Funds Disbursement Agent**: Controls the timing and release of approved loan funds.
- **Debt Management Agent**: Adjusts repayment amounts and debt terms based on borrower status.

Individuals cycle through loan-related states: applying, awaiting funds, and repaying or defaulting on loans. Loan repayment improves borrowers' financial profiles, while defaults have negative effects. Borrowers may re-enter the applicant pool, reflecting recurring financial needs and credit cycles.

***MAFE-Edu.*** This environment models educational and labor market dynamics in a population transitioning between schooling and employment, guided by four decision-making agents:

- **University Admissions Agent**: Selects individuals for university enrollment.
- **University Budget Agent**: Allocates institutional funding, impacting resource quality and student outcomes.
- **Employer Agent**: Sets workforce salaries based on qualifications and degree attainment.
- **Central Planner**: Invests in tertiary education, university infrastructure, and workforce equity initiatives.

Individuals move from a tertiary education pool into university or directly into the workforce. Students may drop out or graduate, with degree duration influencing job prospects. Educational outcomes affect salary offers, linking academic achievement to economic mobility.

**MARL Compatibility.** Our MAFEs follow the standard MARL API, using `step()` and `reset()` methods and returning `observations`, `dones` flags, and `rewards` that include both reward and fairness component vectors. However, realistic fairness-aware modeling introduces structural challenges:

- Observations may include variable-length entity sets (e.g., hospital patients), requiring permutation-equivariant architectures.
- Agents must process structured, high-dimensional outputs from reward and fairness component functions.

These settings remain compatible with MARL libraries such as PettingZoo and EPyMARL, but benefit from specialized architectures—such as DeepSets or GNNs—for effective policy learning, making MAFEs the **first permutation equivariant environment**, a key previously unexplored choice. We provide example environments and reference implementations to support development. These environments provide a flexible testbed for studying fairness-aware decision-making. In the next section, we formalize a cooperative multi-agent use case of our environments and describe how reward and fairness metrics are constructed from environment trajectories.

## 5 A Use Case in Fairness-Aware MARL

**Setting and Objective.** To demonstrate how MAFE can guide fairness-aware decision-making, we define a cooperative multi-agent setting where all agents share a global objective function incorporating both utility and equity.

Table 1: Reward and Fairness Metric in MAFE Instantiations

| Environment | Reward Metrics ($R^{(i)}$) | Fairness Metrics ($F^{(j)}$) |
|---|---|---|
| MAFE-Health | $R^{(1)}$: Insurance profits
$R^{(2)}$: Global negative mortality rate
$R^{(3)}$: Global insured rate | $F^{(1)}$: Mortality rate disparities across regions
$F^{(2)}$: Insured rates disparities across regions |
| MAFE-Loan | $R^{(1)}$: Bank profits
$R^{(2)}$: Global admissions rates
$R^{(3)}$: Global negative default rate | $F^{(1)}$: Admissions rate disparities b/w groups
$F^{(2)}$: Loan wait time disparities b/w groups
$F^{(3)}$: Disparities in default rates b/w groups |
| MAFE-Edu | $R^{(1)}$: Employer profits
$R^{(2)}$: Global university admissions rate
$R^{(3)}$: Global graduation rate | $F^{(1)}$: Admissions rate disparities b/w groups
$F^{(2)}$: Graduation rate disparities b/w groups
$F^{(3)}$: Average salary disparities b/w groups |

Let $o_{n,t}$ and $a_{n,t}$ denote the observation and action of agent $n$ at time $t$. Define the joint histories $o_{1:T}, a_{1:T}$ and consider $K$ reward components $R_n^{(k)}$ and $M$ fairness metrics $F_n^{(m)}$ computed from component functions $c_n^{(R)}$ and $c_n^{(F)}$. $\theta_n$ represent the parameters of the model used to produce the action taken by agent $n$. $\alpha_k$ and $\beta_m$ are user-defined weights for the $k^{th}$ reward and the $m^{th}$ fairness penalty respectively. In the cooperative setting, all agents share the same objective, meaning that $R_n^{(k)} = R^{(k)}$, $F_n^{(m)} = F^{(m)}$, and weights, yielding the following optimization problem for each agent $n$:

$$\max_{\theta_n} \quad \sum_{k=1}^{K} \alpha_k \mathbb{E}_{\theta_n}[R^{(k)}] + \sum_{m=1}^{M} \beta_m \mathbb{E}_{\theta_n}[F^{(m)}]. \tag{1}$$

**Metric Construction from Component Functions.** At each time step, the reward and fairness component functions $c^{(R)}$ and $c^{(F)}$ emit vectors $\mathbf{r}_t$ and $\mathbf{f}_t$ capturing primitive quantities (e.g., profits, admissions, outcomes by group). The final metrics $R^{(k)}$ and $F^{(m)}$ are computed from these quantities via aggregation across time and, in fairness metrics, across groups.

*Reward Structures.* We implement two forms of reward metrics:

- **Aggregated Direct Rewards**: The sum of scalar values across time. For example, in MAFE-Health, total insurance profit is computed by summing per-time-step profits over an episode. For simplicity, we henceforth refer to this type of rewards as *direct rewards*.
- **Ratio-after-aggregation Rewards**: A ratio of two aggregated quantities. For example, the episode-level mortality rate is computed as the total number of deaths divided by the total population observed over time. For simplicity, we henceforth refer to this type of rewards as *rate-based rewards*.

*Fairness Structures.* Fairness metrics quantify disparities in outcome rates across sensitive groups and differ based on group count:

- **Two-group Disparity**: The absolute difference in ratio-after-aggregation statistics between two groups (e.g., minority vs. majority graduation rates in MAFE-Edu).
- **D-group Disparity**: The standard deviation of the ratio-after-aggregation statistics across $D > 2$ groups (e.g., mortality rates across geographic regions in MAFE-Health).

We provide instantiations of reward and fairness metrics used in each of our MAFEs in Table 1. For a comprehensive overview of the full metric definitions, see Appendix B. Our work focuses on group-based disparities, among the most widely used notions in prior FairAI literature Grupen et al. (2022) and align well with real-world policy frameworks in the domains we model (e.g., disparities in mortality or credit access).

## 6  Results and Analysis

In Sections 6.1 and 6.2, we focus on MAFE-Health to illustrate how interventions mitigate disparities and how agents learn under fairness-aware objectives.

To support these analyses, we introduce the *Fair Multi-Agent Cross Entropy Method (F-MACEM)*. The F-MACEM is an extension of the standard cross-entropy method (CEM), tailored to multi-agent systems with fairness considerations. The standard CEM is an evolutionary policy-based algorithm that optimizes a policy by sampling its parameters from a parametric distribution, such as a Gaussian. For each sample, the policy weights, $\theta$, are used to run a full episode, and the resulting rewards are observed. In each training epoch, multiple episodes are run with different policy weight samples. The top-performing policies, referred to as the elite set, are then used to update the distribution from which the policy weights are sampled. This process iterates until the average episodic rewards converge. We use CEMs as a representative and widely adopted evolutionary algorithm, while noting that several alternative formulations exist Szita & Lorincz (2008); Banks et al. (2023); Amaya et al. (2021). In the fully cooperative MARL setting, the standard CEM can be directly extended to handle multiple agents by updating the model weights for all $N$ agents, $\boldsymbol{\theta} = \{\theta_1, ..., \theta_N\}$, simultaneously in each epoch. This update is based on the top-performing weight samples, which maximize episode rewards. These elite samples are then used to update the distribution from which $\boldsymbol{\theta}$ is drawn. An overview of the algorithm is provided in Algorithm 1.

For completeness, Appendix D presents the broader experimental suite. This includes (1) the same disparity-mitigation and learnability experiments for `MAFE-Loan` and `MAFE-Edu`, (2) an ablation on the number of learnable agents to assess the utility of multi-agent versus single-agent learning, (3) Pareto frontiers that characterize the fairness–reward tradeoff, (4) an action analysis examining the impact of different agent strategies, and (5) a comparison of F-MACEM with fairness-augmented policy gradient baselines (`F-MADDPG`, `F-MAPPO`) in the `MAFE-Loan`. Together, these extended results confirm that MAFEs provide a versatile testbed for fairness-aware multi-agent learning.

## 6.1 Validating Interventions for Correcting Disparities

This section shows that actions shaped in our `MAFE-Health` environment can effectively mitigate disparities. Each MAFE is designed to incorporate structural biases, which may lead to disparate outcomes across demographic groups. In the healthcare setting, the core attribute influencing outcomes is the health risk score, which reflects inherent biases across sensitive groups. These scores are calculated by regressing over dataset features used to construct the population, and to support fairness research, we further resample the original feature distributions to exacerbate disparities. Figure 3a illustrates these biased distributions at the start of each episode.

To assess whether agent actions can correct disparities, we conducted fixed intervention experiments in `MAFE-Health`, summarized in Figures 3b–3d. These experiments validate that each intervention has the expected causal effect in isolation, without confounding from interacting adaptive policies. For example, increasing hospital bed availability should reduce mortality; if we instead used adaptive agents that jointly optimize premiums, subsidies, and infrastructure investment, it would be impossible to attribute changes to a single factor. These fixed-action experiments serve as environment validation, ensuring that adaptive algorithms in Sec. 6.2 operate over meaningful, well-calibrated dynamics.

Using a fixed random seed, we compare outcomes in environmental indicators with and without targeted interventions, repeating the process across five seeds. We evaluated incidence and mortality rates under varying conditions such as hospital bed availability, insurance coverage, and public health investments.

The results in these figures illustrate significant improvements when interventions are applied (dashed red lines) compared to baseline scenarios (solid black lines). The direction of the arrow (upward or downward) above each plot signifies improvement in the indicator of interest, indicating the positive impacts that these interventions have on population outcomes. Thus, applying such interventions strategically for sub-population groups allows agents to effectively mitigate disparities across sensitive attributes.

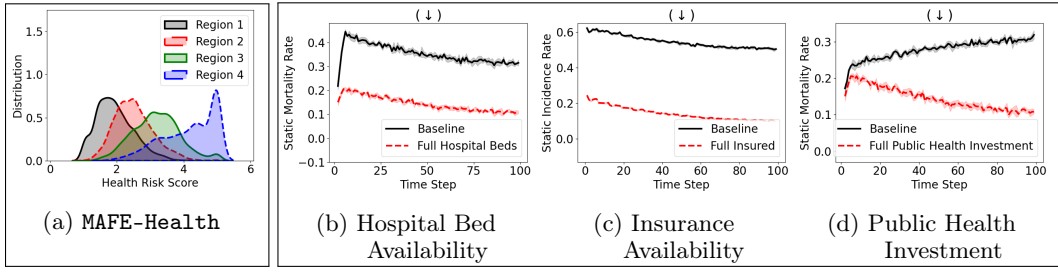

Figure 3: (a) Distribution plots that illustrate disparities in health risk score distributions among geographic sub-populations in MAFE-Health. (b)–(d) Impact of providing hospital beds, universal health insurance, and unlimited public health investment on mortality rates in MAFE-Health. The baseline curves represent the system's outcomes when, all else being equal, the intervention being studied is not applied at all. Shaded regions provide standard deviations over random seeds.

For completeness, analogous experiments for MAFE-Loan and MAFE-Edu are provided in Appendix D.1, where we observe qualitatively consistent trends: targeted interventions improve outcomes and reduce disparities across demographic groups.

## 6.2 COMPOUND EFFECTS OF REWARD TERMS

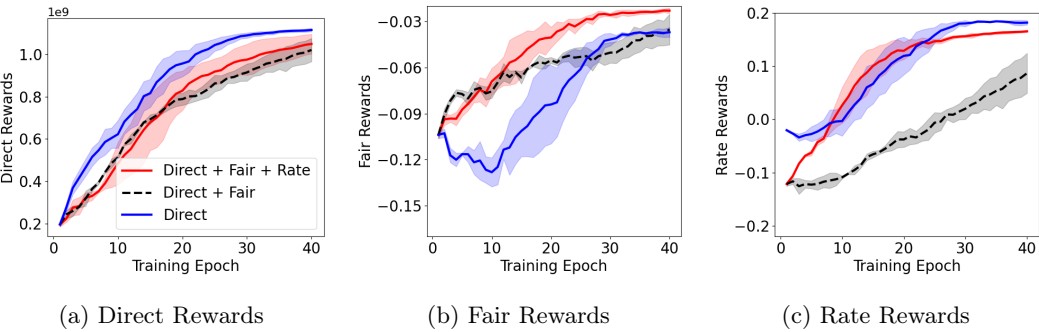

Figure 4: Learning curves for MAFE-Health showing realized rewards obtained during training for models with different combinations of reward terms explicitly included in the F-MACEM's objective function: "Direct"; "Direct + Fair"; or "Direct+Fair+Rate" in the objective. Shaded regions provide standard deviations over random seeds.

In this section, we explore the cumulative impact of incorporating different terms into the F-MACEM's objective function within MAFE-Health, specifically examining how various combinations of terms influence the observed outcomes. We categorize these terms into three groups, as outlined in Section 5: direct rewards, fairness penalties, and rate-based rewards. To analyze their effects, we train F-MACEM using three configurations of the objective: (1) including only direct rewards, (2) including both direct rewards and fairness penalties, and (3) including direct rewards, fairness penalties, and rate-based rewards. For consistency, all elements in each configuration are uniformly weighted.

The results of this analysis for MAFE-Health are presented in Figure 4. Each sub-figure tracks the evolution of a specific reward category throughout training. Within each plot, the plotted curves differentiate the explicit reward terms included in the objective function. As expected, the red line—representing the objective function that incorporates all reward categories— shows steady improvement across all reward types during training. In contrast, configurations excluding certain terms often exhibit less consistent and more volatile performance.

Notably, MAFE-Health shows smaller performance differences between training configurations than observed in other environments. This reflects its design: individuals transition between

healthy, sick, and deceased states, with insurance profit as the primary reward. Insurers benefit most when the population maintains a high insured rate and remains healthy, minimizing claims. As a result, agents learn to balance interventions that optimize both profitability and health outcomes. This alignment between agent objectives and system well-being offers a key insight: even when explicit stakeholder priorities diverge, overlapping indirect objectives can foster cooperative strategies that outperform narrow, self-serving approaches.

For completeness, analogous experiments in `MAFE-Loan` and `MAFE-Edu` are included in Appendix D.2, where we observe qualitatively similar patterns. In those settings, excluding certain reward terms leads to sharper performance drops, underscoring the value of integrating diverse reward categories to balance fairness and utility.

## 7 CONCLUSION AND DISCUSSION

In this work, we introduce the concept of Multi-Agent Fair Environments (MAFEs) as a framework for analyzing fairness in multi-agent systems. We provide a formal definition of algorithmic success within a MAFE, and develop three environments—`MAFE-Health`, `MAFE-Loan`, and `MAFE-Edu`—that model key social systems using a Python-based code implementation akin to popular reinforcement learning libraries such as Gym, Gymnasium, and Petting Zoo. Through experimental analysis, we validate that our MAFEs can be used to analyze interventions that correct for system biases.

One key limitation of our work is the focus on cooperative settings across all MAFE analyses, whereas in practice, agents in these systems may have partially or fully conflicting interests. We adopt this cooperative framing to enable consistent comparison across environments, but future work will extend the framework to support competitive and semi-cooperative interactions. Another limitation is the potential for disagreement among domain experts regarding the fidelity of our environment designs. Because human-centric systems are complex and context-dependent, different stakeholders may emphasize different aspects of realism. To address this, we provide detailed documentation of environment mechanics, incorporate data-driven submodule modeling where feasible, and design MAFEs to be modular and easily customizable, enabling researchers to tailor them to a wide range of assumptions and research goals.

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

# Appendix

## Table of Contents

## A  ADDITIONAL RELATED WORK

**Agent-based Social Simulations.** Agent-based models (ABMs) have been employed to study various societal phenomena, such as the spread of misinformation in social networks, the propagation of epidemics, resource management, and economic systems Perez & Dragicevic (2009); Asgharpour et al. (2010); Giabbanelli et al. (2021); Benthall et al. (2021); Gausen et al. (2022). ABMs offer a bottom-up approach to understanding sociological phenomena, where the interactions between individual agents can lead to emergent behaviors Elsenbroich & Polhill (2023). Traditionally, such modeling has been conducted using surveys, network analysis, data mining, and game theory Bonabeau (2002). Recently, MARL has emerged as a powerful tool for analyzing complex group dynamics Busoniu et al. (2008). However, the majority of existing MARL environments focus on specialized applications, such as games or autonomous navigation Terry et al. (2021); Li et al. (2022) with limited relevance to fairness-oriented research. In contrast, our work analyzes fairness—an essential metric for assessing social and institutional interactions—in an MARL context.

## B  REWARD AND FAIRNESS METRIC DEFINITIONS

In this section, we define the specific reward and fairness structures used in our cooperative use case. While MAFEs support arbitrary composite functions, the examples presented here focus on two common structures: direct and rate-based rewards, and group disparity metrics. These serve to illustrate the expressiveness of our framework and provide interpretable measures in the Healthcare, Loan, and Education MAFEs.

### B.1  REWARD STRUCTURE CUSTOMIZATION

We design two types of rewards for agents: **direct** rewards and **rate-based** rewards. Direct rewards are explicit values, such as profits, that an agent aims to optimize. Rate-based rewards are expressed as ratios, such as the proportion of insured individuals to the total population, representing relative measures that agents aim to optimize. With this, we now provide the form of the reward summation in Problem 1.

Let $K = j + l$, and define the reward components $[r_{1,t}, ..., r_{j+2l,t}] = c^{(R)}(\mathbf{o}_{1:\infty}, \mathbf{a}_{1:\infty})$, where $r_{1,t}, ..., r_{j,t}$ are the direct rewards, $r_{j+1,t}, ..., r_{j+l,t}$ are numerators for rate-based rewards, and $r_{j+l+1,t}, ..., r_{j+2l,t}$ are denominators for the rate-based rewards at time $t$. Then, the final structure of the rewards summation in Equation 1 can be rewritten as the sum of its direct and rate-based constituents:

$$\sum_{i=1}^{j} \alpha_i \left[ \sum_{t=0}^{\infty} \gamma^t r_{i,t} \right] + \sum_{i=j+1}^{j+l} \alpha_i \left[ \frac{\sum_{t=0}^{\infty} \gamma^t r_{i,t}}{\sum_{t=0}^{\infty} \gamma^t r_{i+l,t}} \right]. \tag{2}$$

A concrete example of Equation (2) can be found in the healthcare MAFE description provided in Appendix G. For this environment, the direct reward term (the first summation in Equation (2)) corresponds to a single reward type: insurance profits. The rate-based reward term (the second summation) includes three types of rate-based measures: insured rates, negative incidence rates, and negative mortality rates.

### B.2  FAIRNESS MEASURE STRUCTURE CUSTOMIZATION

Given that the most common disparities in algorithmic fairness are rate-based, such as differences in insured rates across geographic regions in healthcare, we now describe how $F^{(m)}$ in Problem 1 is structured to measure these disparities when the number of groups is two or more.

**Two-group case.** In the two-group case, the disparity between two groups is measured using the directly interpretable absolute difference in rates. Define the fairness components $[f_{1,t}, ..., f_{4M,t}] = c^{(F)}(\mathbf{o}_{1:\infty}, \mathbf{a}_{1:\infty})$, where $f_{4m-3,t}, ..., f_{4m,t}$ represent the numerator and denominator for the rates of Groups 1 and 2 for the $m^{th}$ fairness measure. Then, the fairness

---

**Algorithm 1** Fair Multi-Agent Cross Entropy Method (F-MACEM)

---

1: **repeat**
2:     Initialize buffers $\mathcal{R}$ and $\mathcal{P}$ and parameters $\boldsymbol{\mu}$ and $\boldsymbol{\sigma}^2$
3:     **for** episode = 1... number-of-episodes **do**
4:         Sample $\boldsymbol{\theta} = \{\theta_1, \ldots, \theta_N\}$ from $\mathcal{N}(\boldsymbol{\mu}, \mathrm{diag}(\boldsymbol{\sigma}^2))$
5:         Run episode, storing rewards and fairness components in $\mathcal{R}$ and $\boldsymbol{\theta}$ in $\mathcal{P}$
6:     **end for**
7:     Update $\boldsymbol{\mu}$ and $\boldsymbol{\sigma}^2$ based on top $p\%$ of policies ranked by: $\sum_{k=1}^{K} \alpha_k R^{(k)} + \sum_{m=1}^{M} \beta_m F^{(m)}$
8: **until** Convergence
9: **Return** $\boldsymbol{\theta} = \boldsymbol{\mu}$

---

violation is given by:

$$F^{(m)} = -\left| \frac{\sum_{t=0}^{\infty} \gamma^t f_{4m-3,t}}{\sum_{t=0}^{\infty} \gamma^t f_{4m-2,t}} - \frac{\sum_{t=0}^{\infty} \gamma^t f_{4m-1,t}}{\sum_{t=0}^{\infty} \gamma^t f_{4m,t}} \right| \qquad (3)$$

Both the Loan and Education MAFEs in Appendices F and H provide examples of the two-group sensitive attribute. In each environment, the sensitive attribute identifies whether a person belongs to a minority or majority demographic group. In the Loan MAFE disparities may arise between these groups with respect to key financial indicators, including admissions rates, average wait times, and and default rates. In the Education MAFE disparities may arise between these groups with respect to educational and career indicators, including university admissions rates, graduation rates, and average salaries.

***D*-group case.** When the number of groups, $D$, exceeds two, an absolute difference is inadequate for capturing disparities, as it fails to reflect the distribution of rates across multiple groups. To address this, we use standard deviation to quantify fairness disparities in the $D$-group case. Its simplicity provides an interpretable measure of how evenly rates are distributed among groups, making it particularly suitable for assessing fairness in multi-group settings. We define this measure as follows. Let the fairness components, $[f_{1,t}, ..., f_{2DM,t}] = c^{(F)}(\mathbf{o}_{1:\infty}, \mathbf{a}_{1:\infty})$, where $f_{2D(m-1)+1,t}, ..., f_{2Dm,t}$, provide the numerator and denominator of each of $D$ groups for which we use for measuring the $m^{th}$ rate. Let $Y_d^{(m)} = \frac{\sum_{t=0}^{\infty} \gamma^t f_{2D(m-1)+d,t}}{\sum_{t=0}^{\infty} \gamma^t f_{2D(m-1)+d+1,t}}$ and $\mu^{(m)} = \frac{1}{D} \sum_{d=1}^{D} Y_d^{(m)}$. Then, the fairness measure is given by:

$$F^{(m)} = -\sqrt{\frac{\sum_{d=1}^{D} \left( Y_d^{(m)} - \mu^{(m)} \right)^2}{D}} \qquad (4)$$

As the value of $F^{(m)}$ approaches its upper limit of 0, the disparity in rates across different demographic groups diminishes, improving the parity among them.

An example of a $D$-group sensitive attribute appears in the Healthcare MAFE in Appendix G. In this environment, geography serves as the sensitive attribute, and disparities may arise across four different geographic regions with respect to key health indicators, including mortality rates, incidence rates, and insured rates.

## C  A MULTI-AGENT ALGORITHM FOR SOLVING A MAFE

In this section, we introduce the **Fair Multi-agent Cross Entropy Method (F-MACEM)**, a simple yet effective algorithm for optimizing the objective function in Problem 1. The F-MACEM is an extension of the standard cross-entropy method (CEM), tailored to multi-agent systems with fairness considerations. This method is employed for performance analysis in Section 6.

The standard CEM is an evolutionary policy-based algorithm that optimizes a policy by sampling its parameters from a parametric distribution, such as a Gaussian. For each sample, the policy weights, $\theta$, are used to run a full episode, and the resulting rewards are observed. In each training epoch, multiple episodes are run with different policy weight samples. The

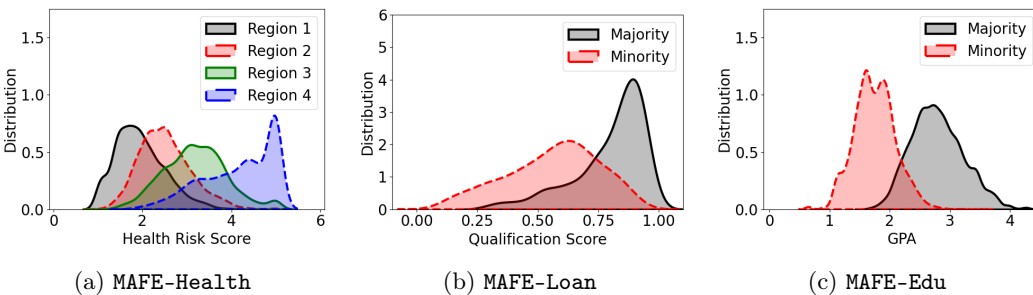

(a) MAFE-Health        (b) MAFE-Loan        (c) MAFE-Edu

Figure 5: Distribution plots that illustrate disparities in (a) health risk score distributions among geographic sub-populations in MAFE-Health, (b) the qualification score distributions of customers in MAFE-Loan, and (c) GPA score distributions of students in MAFE-Edu at the beginning of an episode.

top-performing policies, referred to as the elite set, are then used to update the distribution from which the policy weights are sampled. This process iterates until the average episodic rewards converge.

In the fully cooperative MARL setting, the standard CEM can be directly extended to handle multiple agents by updating the model weights for all $N$ agents, $\boldsymbol{\theta} = \{\theta_1, ..., \theta_N\}$, simultaneously in each epoch. This update is based on the top-performing weight samples, which maximize episode rewards. These elite samples are then used to update the distribution from which $\boldsymbol{\theta}$ is drawn. An overview of the algorithm is provided in Algorithm 1.

# D ADDITIONAL EXPERIMENTS

## D.1 VALIDATING INTERVENTIONS FOR CORRECTING DISPARITIES (UNABRIDGED)

This appendix section provides the unabridged version of Section 6.1 from the main body, including the MAFE-Health, MAFE-Loan, and MAFE-Edu results. Each MAFE is designed to incorporate structural biases, which may lead to disparate outcomes across demographic groups. The core attributes influencing outcomes vary by environment: health risk scores in the MAFE-Health, qualification scores in MAFE-Loan, and baseline GPA in the MAFE-Edu. These attributes reflect inherent biases across sensitive groups, calculated by regressing over dataset features used to construct each MAFE's feature vectors. To enhance these biases for the purpose of supporting fairness research, we have resampled the original feature distributions, exacerbating disparities. Figure 5 illustrates these biased distributions at the start of each MAFE episode.

To assess whether agent actions can correct disparities, we conducted fixed intervention experiments, summarized in Figure 6. Using a fixed random seed, we compare outcomes in environmental indicators with and without targeted interventions, repeating the process in five seeds. In MAFE-Health, we evaluated incidence and mortality rates under varying conditions such as hospital bed availability, insurance coverage, and public health investments. In MAFE-Loan, we examined debt management's effect on qualification scores. In MAFE-Edu, we analyzed the impact of investments, scholarships, mentorship programs, and diversity incentives on graduation rates and employer utility.

The results shown in Figure 6 illustrate significant improvements when interventions are applied (dashed red lines) compared to baseline scenarios (solid black lines). In each plot, there is significant bias in the red dash line when compared with the black solid lines. The direction of the arrow (upward or downward) above each plot signifies improvement in the indicator of interest, indicating the positive impacts that each intervention has on improving outcomes for members of the population. Thus, applying these interventions strategically for sub-population groups should allow agents to effectively mitigate disparities among different sensitive attribute groups.

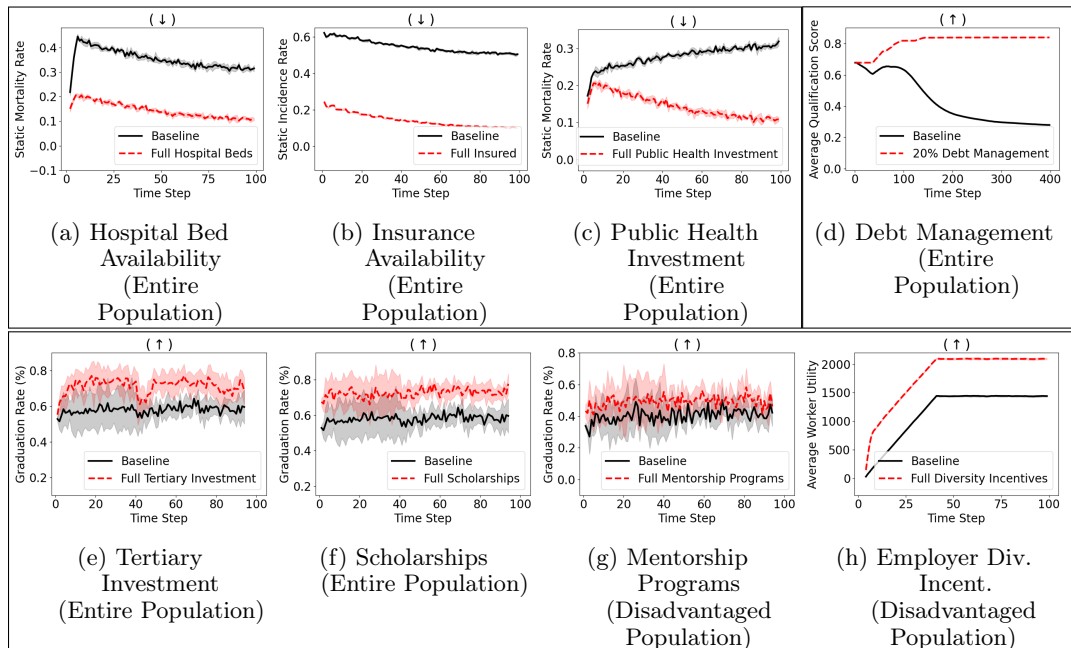

Figure 6: Impact of various interventions in each environment, isolating their effects while holding other factors constant. (a)–(c) In `MAFE-Health`, the effects of providing hospital beds, universal health insurance, and unlimited public health investment on mortality rates. (d) In `MAFE-Loan`, the effect of 20% debt relief on qualification scores for the full population. (e)–(g) In `MAFE-Edu`, the effects of unlimited tertiary investment, full scholarships, and mentorship on graduation rates for the full population and the disadvantaged population. (h) In `MAFE-Edu`, the effect of unlimited diversity incentives for the Employer Agent on the average utility of workers from disadvantaged groups. Shaded regions provide standard deviations over random seeds.

## D.2    COMPOUND EFFECTS OF REWARD TERMS (UNABRIDGED)

This appendix section provides the unabridged version of Section 6.2 from the main body, including the `MAFE-Health`, `MAFE-Loan`, and `MAFE-Edu` results. We particularly explore the cumulative impact of incorporating different terms into the F-MACEM's objective function for each MAFE, specifically examining how various combinations of terms influence the observed outcomes for each individual term. We categorize these terms into three distinct groups, as outlined in Section 5: direct rewards, fairness penalties, and rate-based rewards. To analyze their effects, we train the F-MACEM using three configurations of the objective function: (1) including only direct rewards, (2) including both direct rewards and fairness penalties, and (3) including direct rewards, fairness penalties, and rate-based rewards. For consistency, all elements in each objective function are uniformly weighted.

The results of this analysis are presented in Figure 7. Each row corresponds to a different environment, while each column tracks the evolution of a specific reward category throughout training. Within each plot, the plotted curves differentiate the explicit reward terms included in the objective function. As expected, the red line—representing the objective function that explicitly incorporates all reward categories—shows steady improvement across all reward types during training. In contrast, configurations excluding certain terms often exhibit less consistent and volatile performance. For example, in `MAFE-Edu`, the rate-based reward curve for the F-MACEM, trained solely with direct rewards, declines from its initial value during training and only approximately returns to its starting point by the final epoch on average. Similarly, in `MAFE-Loan`, excluding rate-based rewards causes the corresponding reward curve to plateau at a significantly lower value than observed in the fully-inclusive configuration.

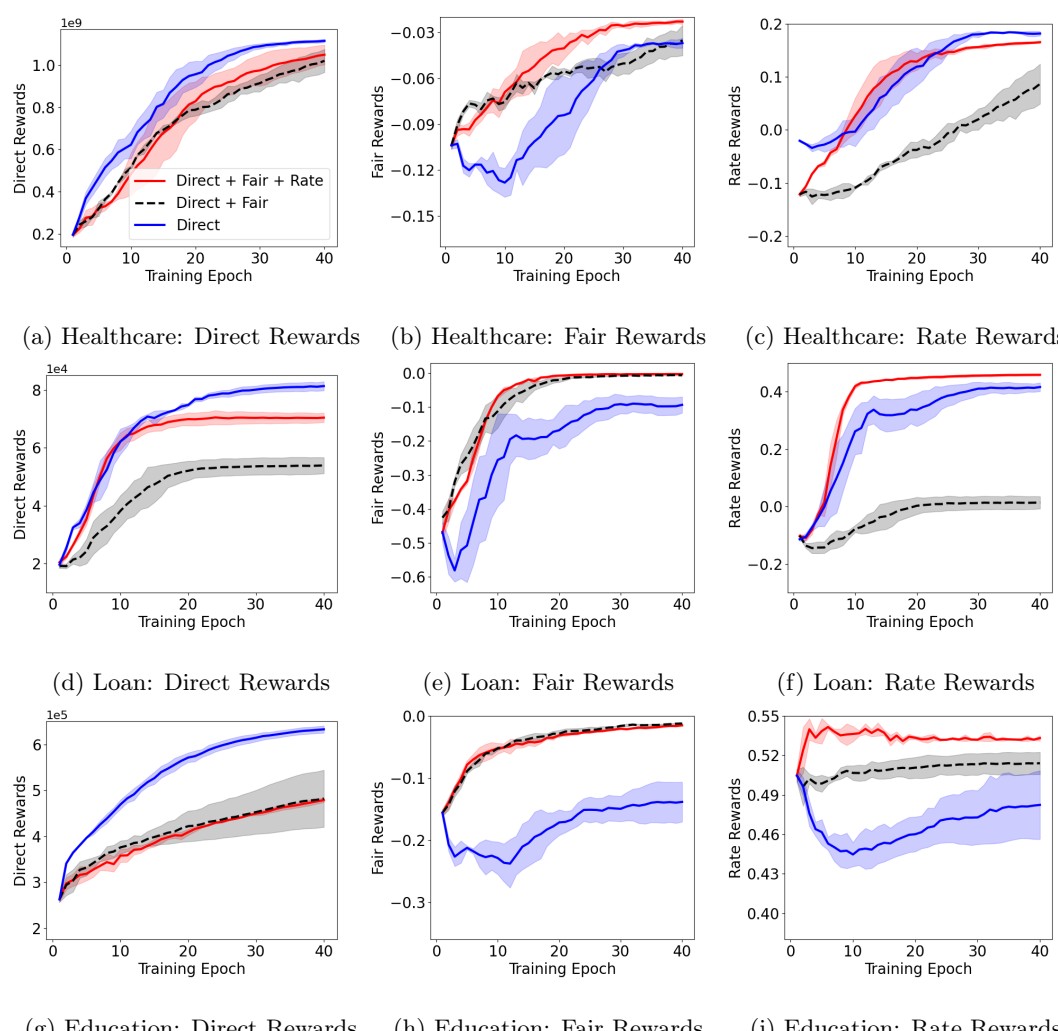

(a) Healthcare: Direct Rewards     (b) Healthcare: Fair Rewards     (c) Healthcare: Rate Rewards

(d) Loan: Direct Rewards     (e) Loan: Fair Rewards     (f) Loan: Rate Rewards

(g) Education: Direct Rewards     (h) Education: Fair Rewards     (i) Education: Rate Rewards

Figure 7: Learning curves showing realized rewards obtained during training for models with different combinations of reward terms explicitly included in the F-MACEM's objective function: "Direct"; "Direct + Fair"; or "Direct+Fair+Rate" in the objective. Shaded regions provide standard deviations over random seeds.

These patterns underscore the utility of integrating diverse reward terms to balance learning objectives effectively within each MAFE.

This analysis also highlights environment-specific characteristics. Notably, `MAFE-Health` shows smaller performance differences between training configurations compared to `MAFE-Loan` and `MAFE-Edu`. While this might seem counterintuitive, it reflects the MAFE's design: individuals transition between healthy, sick, and deceased states, with insurance profit as the primary reward. Insurers benefit most when the population maintains a high insured rate and remains healthy, minimizing claims. As a result, agents learn to balance interventions that optimize profitability and health outcomes. This alignment between agent objectives and system well-being offers a key insight: even when explicit stakeholder priorities diverge, overlapping indirect objectives can foster cooperative strategies that outperform narrow, self-serving approaches.

## D.3 ASSESSING THE BENEFIT OF MULTI-AGENT LEARNING

In this section, we perform an experiment to assess the benefits of allowing multiple agents to learn dynamic policies, using the `MAFE-Loan` as a testbed. Specifically, we compare the performance of multi-agent learning, where all agents are allowed to learn optimal policies, against single-agent learning scenarios and a fixed policy baseline. The optimal policy, in this case, is defined as the one that maximizes the Loan MAFE's objective function (as defined in equation 1), with uniform weighting applied to all terms in the objective.

We begin by establishing a baseline with a fixed policy. In this scenario, the system consists of three agents: the Admissions and Debt Management Agents, each producing two actions—setting an admissions threshold and a debt management factor for each of the binary demographic groups—and the Disbursement Agent, which generates a scoring vector for the individuals in the loan queue. The fixed policy is generated by randomly sorting the individuals in the queue, which leads to equal average wait times across demographic groups.

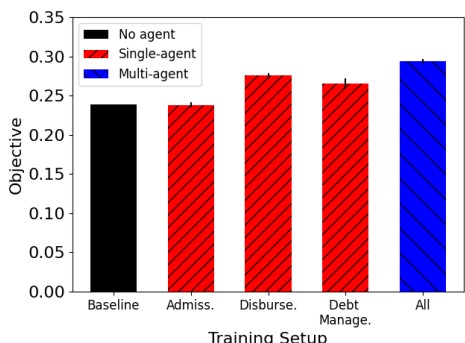

Figure 8: Performance for the baseline fixed policy, single-agent learning (one agent learns dynamically), and multi-agent learning (all agents learn dynamically). Higher values indicate better performance.

Next, we identify the actions for the Admissions and Debt Management Agents through a two-tier grid search to optimize the objective function. In the first tier, we search for the best global pair of admissions threshold and debt management factor by partitioning the action space over the $[0, 1]$ interval. Here, "global" means the same pair of values is applied to both demographic groups. In the second tier, we perform a grid search to determine how much to deviate the group-specific values from the global values, resulting in optimal values of $[0.0, 0.0]$ for admissions thresholds and $[0.12, 0.18]$ for debt management factors, where the first value corresponds to the advantaged group and the second to the disadvantaged group.

Once the baseline fixed policy is established, we conduct three forms of single-agent training sessions. In each, one of the agents is trained while the other two agents are fixed according to the baseline policy.

The results comparing the fixed policy, single-agent training, and multi-agent training are shown in Figure 8. The plots display the resulting values of the objective function for each policy implementation, with higher values indicating better performance in maximizing the objective. Since the fixed policy was optimized to perform well according to the objective function, its performance is relatively high. However, allowing agents to learn, rather than relying on fixed or heuristic policies, leads to further improvements in agent performance. In particular, the multi-agent training scenario achieves the highest performance, demonstrating the utility of multi-agent learning in environments with multiple decision points. This underscores the value of considering multi-agent interactions, rather than simplifying the system to a single decision point with heuristic approaches.

### D.4 Reward-Fairness Frontier in MAFEs

In this section, we analyze the F-MACEM algorithm's performance in achieving fairness and accuracy, measured by the reward and fairness terms in Equation 1. Particularly, each reward and fairness violation is weighted uniformly, with $\alpha_k = \frac{\lambda}{K}$ for rewards and $\beta_m = \frac{1-\lambda}{M}$ for fairness violations. We then train the system using uniformly sampled values of $\lambda$ over the interval $[0, 1]$ to analyze the trade-off between fairness and accuracy. To ensure uniform contribution from each component, we normalize all rewards and fairness violations to lie within the range $[0, 1]$. The normalization factors for these results are provided in Table 8 of Appendix I.

Figure 9 presents the resulting Pareto frontiers, which illustrate the trade-off between accuracy and fairness. Each point on the frontier represents the average performance of

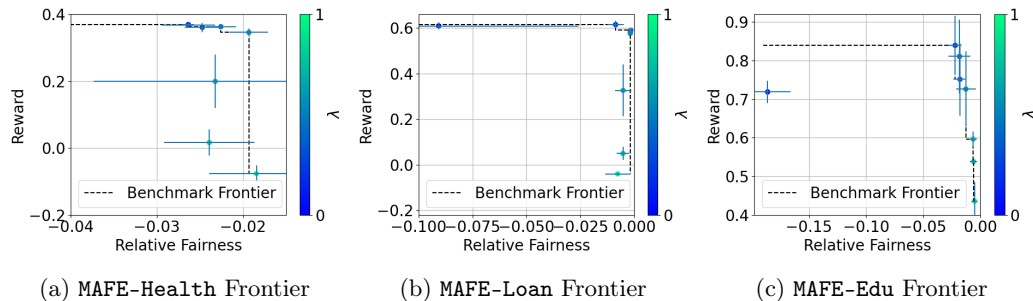

(a) MAFE-Health Frontier     (b) MAFE-Loan Frontier     (c) MAFE-Edu Frontier

Figure 9: Pareto frontiers that demonstrate the reward-fairness tradeoff for the F-MACEM in the (a) MAFE-Health, (b) MAFE-Loan, and (c) MAFE-Health.

a model trained with the same objective function across three different training seeds to represent relative fairness values. Both fairness measures from Equation 3 (for Loan and MAFE-Edu) and Equation 4 (for MAFE-Health) produce negative values, which are plotted directly since they are compatible with maximization. In MAFE-Loan and MAFE-Edu, fairness is assessed using a binary sensitive attribute, with a higher value indicating greater fairness. In contrast, MAFE-Health evaluates fairness across four geographic regions, where a higher value also signifies greater fairness. In all plots, the highest fairness value corresponds to a value of 0.

These results indicate only a subtle trade-off between maximizing rewards and maintaining fairness, with the magnitude of this trade-off varying across different environments. Notably, the most significant performance declines occur when the weight assigned to the fairness term, $1 - \lambda$, substantially exceeds that of the reward term, $\lambda$. However, F-MACEM generally maintains high reward levels when a moderate allowance for fairness violations is incorporated. This robustness suggests that even a small increase in the fairness weight within a reward-centric objective can have a meaningful impact. In particular, disparities can be mitigated over time through effective interventions, and such fairness regularization can, in some cases, improve rewards by helping F-MACEM avoid poor local minima.

D.5    POLICY ACTION ANALYSIS

In this section, we analyze the actions that the F-MACEM learns to produce over the training process when direct rewards, rate-based rewards, and fairness penalties receive uniform weighting in the objective function for each MAFE.

For the MAFE-Loan, we analyze the average admissions threshold set by the Admissions Agent, which determines the number of people approved for loans in an episode, and the debt management factor set by the Debt Management Agent, which helps the customer population avoid loan defaults. In the MAFE-Health, we examine how the Central Planner Agent allocates its budget across interventions and how the Insurance Agent sets premiums. For completeness, we restate the MAFE-Edu action analysis, focusing on how the Central Planner Agent distributes funds for interventions, how the Employer Agent sets salaries, and how the University Budget Allocation Agent allocates resources to improve student academic success.

For the MAFE-Loan, Figure 10a shows the average admission threshold over 40 training epochs. As training progresses, the agent learns to lower the threshold, effectively admitting nearly all applicants. This strategy increases the admission rate among the global population, thereby improving the rate-based reward. However, admitting more applicants without additional safeguards can increase default rates, risking the bank's financial stability. To mitigate this issue, the Debt Management Agent can adjust the debt management factor to aid customers to avoid defaulting. As illustrated in Figure 10b, this agent is able to strategically balance debt adjustment by setting these values neither too high to protect profits, nor too low to avoid widespread defaults. By targeting this aid, the agent ensures similar default rates across both groups, promoting fairness and financial stability.

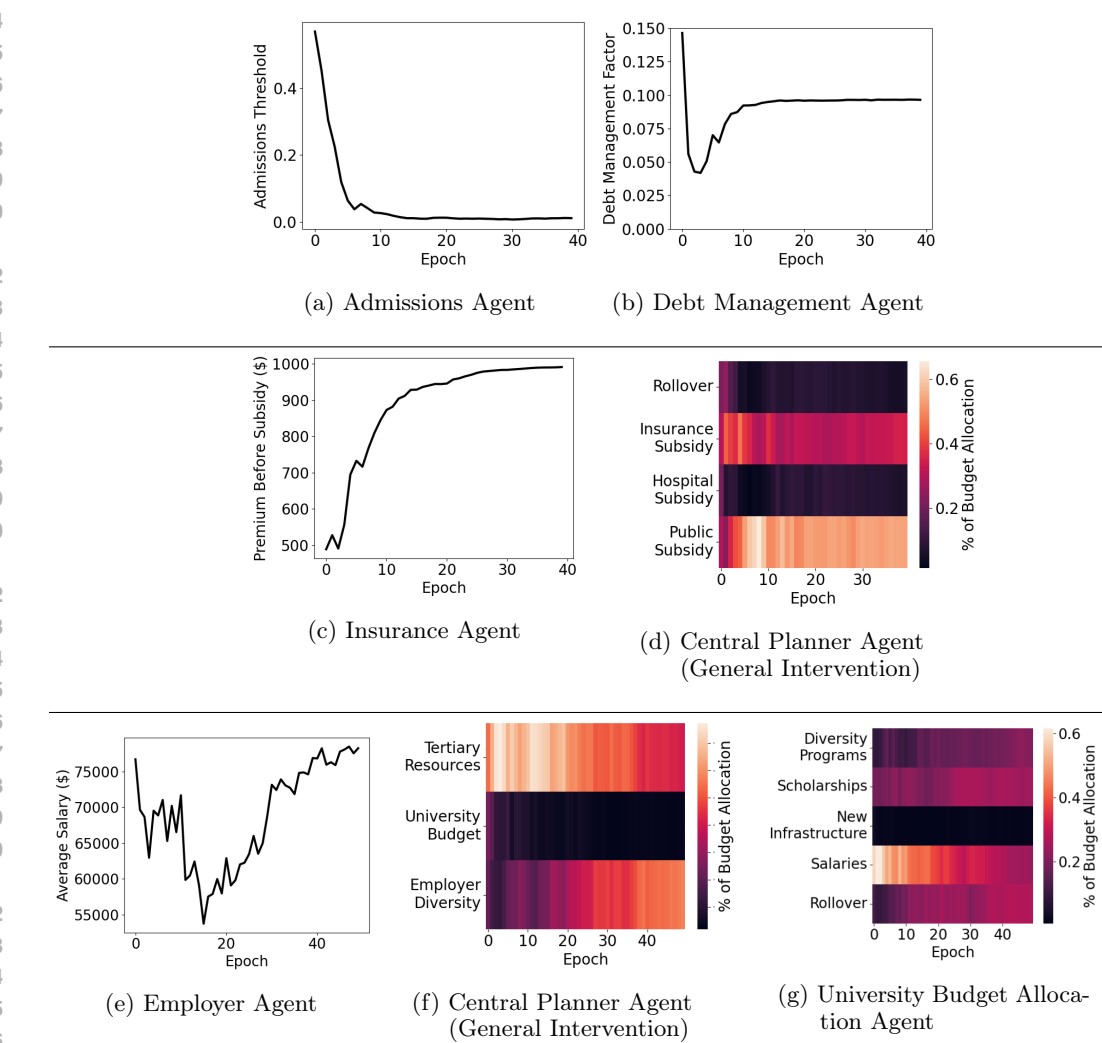

(a) Admissions Agent
(b) Debt Management Agent
(c) Insurance Agent
(d) Central Planner Agent (General Intervention)
(e) Employer Agent
(f) Central Planner Agent (General Intervention)
(g) University Budget Allocation Agent

Figure 10: Average actions taken by agents over training epochs in MAFEs for Loan (Row 1), Healthcare (Row 2), and Education (Row 3).

Figure 10c and 10d present the actions taken by various agents within the `MAFE-Health`. Specifically, Figure 10c highlights the premium-setting behavior of the Insurance Agent. During training, the agent learns to set premiums near the upper limit of $1000. While this might initially seem challenging for affordability, Figure 10d illustrates a heatmap of the average percentage of the Central Planner Agent's budget allocated to healthcare subsidies. The planner prioritizes two main areas: (1) subsidizing insurance premiums to reduce the effective cost for individuals and (2) investing in public health initiatives. These premium subsidies help maintain affordability for consumers, even with the higher nominal premiums. The largest share of the planner's budget is allocated to public health investments, aimed at reducing the overall burden on the healthcare system by preventing illness. This approach focuses on improving baseline health outcomes across the population, complementing reactive measures like treatment subsidies by emphasizing preventive care strategies.

Figure 10e-10g illustrate agent actions in the `MAFE-Edu`. The Central Planner Agent primarily invests in tertiary resources and employer diversity incentives, as shown in Figure 10f, indicating that tuition revenue sufficiently covers university operations. The University Budget Allocation Agent demonstrates an evolving strategy, as shown in Figure 10g. Early in the training process, the agent focuses a significant portion of its budget on faculty salaries to ensure financial stability and avoid potential disruptions. Yet, since faculty salaries in this

MAFE are fixed, the agent recognizes that allocating too large a portion of its resources for them may not be the most efficient use of funds. As the agent refines its strategy, it adjusts its budget distribution, directing more resources toward student-specific interventions, such as scholarships for both majority and underrepresented student groups, as well as mentorship programs for underrepresented groups. This shift in allocation helps address disparities in cumulative GPAs between majority and underrepresented students, ultimately improving educational and career outcomes.

Notably, Figure 10e shows a significant trend reversal in the employer agent's salary-setting behavior midway through the training process. Initially, the employer agent decreases average salaries; however, this trend inverts as training progresses, leading to a steady increase in salaries. This shift results from a combination of factors. First, the Central Planner Agent's investment in diversity incentives directly boosts the salaries of underrepresented minority groups. Second, as the Central Planner and University Budget Allocation Agents optimize their investments in tertiary resources and university student aid, overall student performance improves. These enhancements in educational outcomes translate to better career success, indirectly driving higher salaries.

The coordinated actions among the different agents in each MAFE can create a positive feedback loop for improving various system rewards. Yet the reason this is possible is because the flexible intervention structure that our MAFEs offer.

### D.6 POLICY GRADIENT BASELINES FOR F-MACEM

To evaluate algorithmic performance under our MAFE setup, we examine three fairness-aware methods on the Loan MAFE: our proposed F-MACEM, which uses parameter-space sampling to optimize temporally aggregated fairness and utility objectives, and fairness-augmented variants of Multi-Agent Proximal Policy Optimization and Multi-Agent Deterministic Policy Gradient, denoted F-MAPPO and F-MADDPG, respectively.

These algorithms reflect distinct design assumptions. F-MAPPO and F-MADDPG are adapted from standard policy gradient methods and operate under the assumption that rewards are available as additive, per-time-step signals, using (Gaussian) noise for exploration. In our implementations, fairness and reward components are incorporated directly into the step-wise reward via a weighted combination:

$$r_t = \sum_{k=0}^{K} \alpha_k r_{k,t} + \sum_{m=0}^{M} \beta_m f_{m,t} \tag{5}$$

This formulation supports gradient-based learning by treating all objectives as decomposable over time. In contrast, F-MACEM uses a population-based evolutionary strategy that perturbs policy parameters and evaluates performance over entire trajectories. This allows it to optimize reward structures based on temporally aggregated statistics—such as ratio-after-aggregation fairness metrics used in `MAFE-Loan`. It also enables broader, high-level exploration by sampling from a distribution over full policy parameterizations, rather than relying on local action-space noise.

We compare each algorithm's ability to balance fairness and utility according to the composite objective in Equation 1, where half the total weight is allocated to profit maximization and the remainder is evenly distributed among the fairness metrics listed in Table 1. Results for the full version of `MAFE-Loan` (denoted `MAFE-Loan-F`) are shown in Figure 11a. Notably, F-MACEM achieves significantly better performance on the fairness-reward objective compared to F-MAPPO and F-MADDPG.

To better understand the root of this performance gap, we introduce a simplified variant of `MAFE-Loan` denoted `MAFE-Loan-S`, in which we reduce transition complexity while keeping the action, observation, and reward structures unchanged. This diagnostic setting tests whether the representational structure of our MAFEs is learnable in isolation from their dynamic complexity. If all algorithms perform well on `MAFE-Loan-S`, this suggests that the action-observation-reward interface is compatible with multiple learning paradigms and that the performance issues observed on `MAFE-Loan-F` stem from the difficulty of long-term planning under the more advanced environmental complexity of `MAFE-Loan-F`.

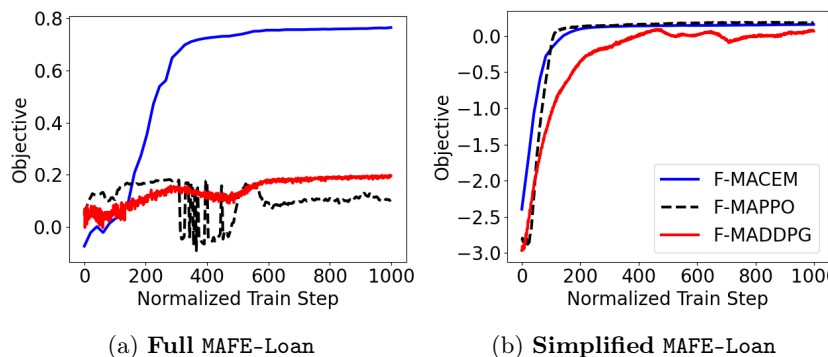

(a) **Full** MAFE-Loan        (b) **Simplified** MAFE-Loan

Figure 11: Performance comparison between F-MACEM, F-MAPPO, and F-MADDPG on MAFE-Loan-F (left) and MAFE-Loan-S (right).

To construct the simplified environment MAFE-Loan-S, we retain the core elements of the full environment (MAFE-Loan-F)—including agent roles, action and observation formats, and the reward/fairness metrics used during training. However, we simplify the environment's internal dynamics to reduce temporal and representational complexity while preserving the fundamental decision-making structure. In MAFE-Loan-F, individuals are tracked across multiple modules (e.g., admissions, disbursement, debt management) over extended time horizons, and system behavior is governed by separate rule-based and statistical models per module. In MAFE-Loan-S, these stages are collapsed into a single-step abstraction per individual. Instead of tracking behavior over dozens of time steps, outcomes are summarized in a single decision event using a unified rule-based transition model. Additionally, the three distinct agent observations are replaced by a single shared observation, population size and feature dimensionality are reduced, and episode length is shortened. Finally, a single logistic regression model is used for both admissions and default prediction, replacing the two distinct models used in the full version. These changes preserve the action-observation-reward interface while significantly reducing planning horizon and state evolution complexity.

Figure 11b confirms that each algorithm exhibits steady learning progress on MAFE-Loan-S, supporting the conclusion that the environment structure is learnable and that algorithm-environment compatibility plays a critical role in overall performance.

**Insights.** These results highlight how differences in exploration strategy and optimization structure can significantly affect learning outcomes in complex multi-agent environments. While all three algorithms share similar learning objectives, F-MACEM's parameter-space exploration enables it to more effectively navigate the long-term dependencies and delayed credit assignment challenges present in MAFE-Loan-F. In contrast, F-MAPPO and F-MADDPG struggle in the full environment, likely due to their reliance on local, action-space noise and assumptions of step-wise reward decomposition. These findings underscore the importance of aligning algorithmic assumptions with environment complexity, particularly when optimizing fairness objectives that depend on temporally aggregated outcomes.

Rather than viewing this as a limitation of our MAFEs, we see it as a call to action for future algorithm development. Instead of simplifying environments to fit existing methods, future work should prioritize designing algorithms capable of engaging with the structural and temporal complexities inherent in real-world fairness-aware decision systems.

## E   COMMON CONSIDERATIONS IN MAFE DESIGN

While each of our MAFEs has unique elements, they also share several common structural characteristics derived from their Fair Dec-POMPs. In this section, we outline the key similarities in their designs.

### E.1 OBSERVATIONS

At a given time step, $t$, Agent $n$ receives an observation $o_{n,t} \subseteq \mathcal{O}_n$. We design the observation space for every agent in each of our environments to take the following form, $\mathcal{O}_n = \{o | o \in \Pi_{m=0}^M \mathbb{R}^{m \times k_n}\}$. Here, $M$ represents the global population size in a given MAFE and $k_n$ denotes the dimensionality of the feature vector associated with each individual containing the features that Agent $n$ can use when deciding on an action.

Moreover, while there may be overlap in the features provided to different agents, this is not guaranteed. As a result, the size of the feature vector $k_n$ varies across agents. For instance, an employer agent may have access to an individual's undergraduate GPA when determining salary offers, but this feature would not be available to a university admissions agent, since high school students do not have an undergraduate GPA.

### E.2 ACTIONS

Agent actions take the general form $\mathcal{A}_n = \{a | a \in \Pi_{m=0}^M \mathbb{R}^m\}$. There are two particular categories of actions that serve as special instances of this structure: (1) **individual-level actions** and (2) **group-level** actions.

For Agent $n$ with observation matrix, $o_{n,t}$, of size $m_{n,t} \times k_n$, an individual-level action takes the form $a_{n,t} \in \mathbb{R}^{m_{n,t}}$. In this case, Agent $n$ produces an action vector, where the $i^{th}$ element corresponds to a decision for the $i^{th}$ individual, whose feature vector is represented by the $i^{th}$ row of $o_{n,t}$. For instance, in MAFE-Health, the Hospital agent could generate an action vector in which each element represents the priority rank assigned to an individual, determining their position in the queue for receiving an available hospital bed.

In contrast, a group-level action affects a subset of individuals in the entire population (subset of the rows of the observation matrix). The structure of a group-level action is $a_{n,t} \in \mathbb{R}^{f_n}$, where $f_n$ represents the number of decisions Agent $n$ must make, which affect all $m_{n,t}$ individuals. For example, in MAFE-Loan, the Debt Management Agent could output a single percentage value that determines the debt adjustment percentage applied to every customer's payment at that time step. In this case the group is the entire customer repayment population.

### E.3 AGENTS

A MAFE is defined as a fair Dec-POMDP, where the decentralization reflects the interaction of $N$ agents with the environment through their respective input actions and output observations, rewards, and fairness components. Specifically, $N$ agents correspond to $N$ distinct input actions provided to the environment and $N$ corresponding output observations, reward component vectors, and fairness component vectors generated by the environment. This decentralization does not necessarily mean that $N$ separate models must be used to generate the actions for each agent, though.

For instance, the $N$ observations, $\{o_{n,t}\}$, could be aggregated into a single global observation, processed by a single AI model, which outputs a unified action vector. This vector can then be split into $N$ individual, actions, $\{a_{n,t}\}$—one for each agent—before being input back into the environment. Alternatively, in a fully decentralized setup, $N$ separate models can process the individual observations independently to generate $N$ actions. A hybrid setup might involve partial aggregation of observations, with subsets of agents sharing models. Thus, while the environment enforces decentralization in terms of interactions with agents, the AI model architecture (centralized, decentralized, or hybrid) remains a design choice and is independent of the underlying MAFE formulation.

However, we require $a_{n,t}$ to be permutation-equivariant with respect to the rows of $o_{n,t}$. For global-level actions, permutation-equivariance ensures that the arbitrary ordering of the rows in an observation does not affect the global decision applied to all individuals influenced by the action. For individual-level actions, permutation-equivariance guarantees that the $i^{th}$ element of the action vector corresponds to the decision for the $i^{th}$ individual in the agent's observation matrix, rather than being associated with any other individual.

### E.4 SENSITIVE ATTRIBUTE

The sensitive attribute refers to the feature for which bias mitigation is necessary, as measured using the binary or $D$-ary metrics defined in Equations 3 and 4 in Section B.2. In `MAFE-Loan` and `MAFE-Edu`, the sensitive attribute is a binary feature indicating whether an individual belongs to an advantaged or disadvantaged group. In `MAFE-Loan`, this could represent attributes such as sex or race, both of which are protected characteristics under U.S. anti-discrimination laws in financial institutions Federal Deposit Insurance Corporation (2021). Similarly, in `MAFE-Edu`, the sensitive attribute reflects whether an individual belongs to an underrepresented minority group at the university level.

In contrast, `MAFE-Health` underscores that much of the disparity in health outcomes across demographic groups is driven by geographic location. For example, families of color—particularly Black families—are more likely to live in areas with limited access to healthcare facilities U.S. Department of Health and Human Services (2024). In this context, geographic location serves as the sensitive attribute, with four distinct regions, each associated with specific health outcome disparities.

### E.5 REWARD AND FAIRNESS COMPONENT FUNCTIONS

In the MAFE framework, the use of component functions for reward and fairness allows for greater flexibility in how these metrics are calculated. Specifically, this design choice enables the calculation of aggregation-based fairness and reward metrics as opposed to step-wise metrics that are computed at each individual time step.

The primary advantage of using component functions rather than directly outputting rewards or fairness values at each time step is that it allows the construction of rate-based terms that aggregate the rewards and fairness violations over time. Directly computing values at each time step would constrain the system to use step-wise measures of fairness (e.g., fairness ratios calculated at each step), which can be sensitive to outliers and fluctuations in the data, as pointed out by Xu et al. Xu et al. (2024). Instead, our approach supports the calculation of aggregation-based metrics, which aggregate over time, offering a more holistic view of fairness across the entire decision-making process.

For example, using step-wise fairness metrics might yield values like:

$$\sum_t^T \frac{\#\text{insured}_t}{\#\text{population}_t} \quad \text{and} \quad \sum_t^T \left| \frac{\#\text{insured}_t^A}{\#\text{population}_t^A} - \frac{\#\text{insured}_t^B}{\#\text{population}_t^B} \right|.$$

While this approach is valid, it only captures fairness at each time step and can be influenced by short-term fluctuations. On the other hand, aggregation-based fairness metrics enable the calculation of measures like:

$$\frac{\sum_t^T \#\text{insured}_t}{\sum_t^T \#\text{population}_t} \quad \text{and} \quad \left| \frac{\sum_t^T \#\text{insured}_t^A}{\sum_t^T \#\text{population}_t^A} - \frac{\sum_t^T \#\text{insured}_t^B}{\sum_t^T \#\text{population}_t^B} \right|.$$

These metrics aggregate relevant quantities across all time steps before computing the fairness ratios, leading to more stable, long-term views of fairness that are less sensitive to the variance at each individual time step.

This flexibility in defining fairness and reward measures provides greater versatility in capturing long-term patterns and overall fairness in decision-making processes, making the MAFE framework adaptable to different applications.

### E.6 TRANSITION FUNCTION

The transition function defines system dynamics, updating the state from time $t$ to $t+1$ based on agent actions. This updated state forms the basis for future observations. While each MAFE's transition function is unique, they all capture complex interactions between

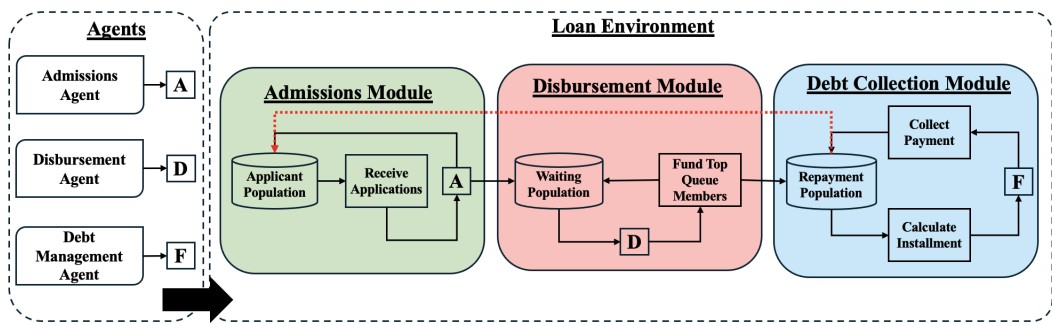

Figure 12: `MAFE-Loan` Diagram

Table 2: `MAFE-Health` Features

| Variable | Origin | How it is updated | Description |
|---|---|---|---|
| RACE | Lending Club | None | Main racial background |
| INTRATE | Lending Club | None | Loan Interest Rate |
| BALANCE | Lending Club | Environment Dynamics | Loan Balance |
| ANNUALINC | Lending Club | Environment Dynamics | Annual income |
| DTI | Lending Club | Environment Dynamics | Debt-to-income ratio |
| FICO_RANGE_LOW | Lending Club | Environment Dynamics | Lower boundary of individual's FICO score range |
| FICO_RANGE_HIGH | Lending Club | Environment Dynamics | Upper boundary of individual's FICO score range |
| TIMETOMATURITY | Environment | Environment Dynamics | Remaining time until loan maturity |
| WARNING | Environment | Environment Dynamics | Flag that loan in danger of default |
| TOTREQUEST | Environment | Environment Dynamics | Total amount requested by bank on current loan |
| TOTRECEIVE | Environment | Environment Dynamics | Total amount received by bank on current loan |
| QUALSCORE | Environment | Environment Dynamics | Qualification score |
| TOTBANKPROF | Environment | Environment Dynamics | Bank's accumulated profits |
| CURRINSTALL | Environment | Debt Agent ($\pi_3$) | Amount of current installment |

agents and individuals, reflecting real-world processes such as loan repayment cycles, health resource allocation, and educational progression.

These state transitions continue until a MAFE episode is terminated. This occurs when one of the following conditions is met:

1. Financial Failure: Entities like an insurance company, employer, or university may go bankrupt after incurring losses that lead to net negative profits or prevent them from paying employees.

2. Terminal Time Step: The episode ends at a user-specified terminal time step.

## F  MAFE-Loan Modeling Details

In this section we provide a detailed explanation of how we design `MAFE-Loan` introduced in Section 4 of the main paper.

**Overview:** A diagram illustrating the design of `MAFE-Loan` is provided in Figure 12. This environment simulates the loan processing pipeline of a financial institution. The agents in this system represent three main branches of the bank. The first is the Admissions Agent ($\boldsymbol{\pi}_1$), responsible for determining who will be approved for loans. The second is the Disbursement Agent ($\boldsymbol{\pi}_2$), which handles the timing of loan disbursements. The third is the Debt Management Agent ($\boldsymbol{\pi}_3$), which oversees loan repayment and manages defaults.

At each time step, a sample of individuals from the applicant population applies for loans. These applicants are either approved or rejected by the Admissions Agent. Rejected applicants are re-entered into the population and may be considered for loans in subsequent time steps. Approved applicants move into the disbursement phase of the loan processing pipeline.

In the disbursement phase, individuals must wait for their loan funds to be disbursed by the institution. The disbursement process is constrained by human resources, meaning only a fixed number of loans can be processed per time step, which may introduce delays. The Disbursement Agent controls who receives their funds first by sorting the queue of individuals waiting for their loans at every time step.

Once an applicant receives a loan, they begin making regular payments in each subsequent time step. If the borrower consistently makes on-time payments until the loan's maturity, the loan is fully paid off. Conversely, if the borrower fails to make timely payments, they will default on the loan. In this phase, the Debt Management Agent has the ability to adjust repayment requests to alleviate financial strain on an individual and help them avoid default.

An individual's features are updated when their loan is terminated, but the nature of the update differs depending on how the loan is terminated: the individual's features improve in the case of successful repayment and deteriorate in the case of default. The individual is then reinserted into the applicant pool to be resampled for future loan applications.

We now elaborate on each entity in the environment by explaining the operations that take place during a given time step, $t$.

**Population:** At the beginning of the loan simulation, a global population is initialized consisting of $N$ individuals. Each individual has an associated feature vector, $\mathbf{v} = [\mathbf{v}_c^T \ \mathbf{v}_v^T]^T \in \mathbb{R}^k$, which contains both financial and demographic attributes used by the agents to make decisions. The vector $\mathbf{v}_c$ represents constant features that remain unchanged throughout the simulation, while $\mathbf{v}_v$ contains variable features that are influenced by the dynamics of the MAFE system.

To ensure that the data used in the simulation is realistic, we leverage real-world data from LendingClub, a financial services company that connects borrowers with investors for peer-to-peer lending Lending Club Dataset. Our population is constructed using loans from this dataset, with initial balances ranging from $1,000 to $40,000. Approximately half of the features in the feature vector are directly derived from the loan data, as outlined in Table 3. These feature vectors are then augmented with additional information relevant to the dynamics of the environment, such as QUALSCORE, which indicates an individual's qualification score and serves as a proxy for the likelihood of loan repayment.

The global population is divided into distinct subpopulations based on the phase of the loan processing system each individual inhabits. These include the **application population**, which consists of individuals not yet in the loan processing system but who wish to apply for loans; the **waiting population**, which includes individuals who have been approved for loans and are awaiting disbursement of funds; and the **repayment population**, which contains individuals who have received their loan funds and are currently repaying them.

The features associated with individuals in each of these categories provide the observations for the various agents involved in the MAFE system, including the Admissions, Disbursement, and Debt Management Agents, at each time step. These features, particularly those in $\mathbf{v}_v$, are influenced by the actions taken by different agents within the system. For example, the bank may adjust an individual's installment plan as they continue to repay their loan. This not only updates the current loan balance (CURRINSTALL), but can also improve or deteriorate financial indicators like DTI and FICO scores over time, depending on the individual's payment behavior. These evolving features provide context to enable the agents

to adjust their strategies to, for example, modify installment amounts to help prevent default or encouraging timely repayments.

In the remainder of this section, we use subscript notation to refer to the value of a particular variable for an arbitrary individual or group at time $t$. For instance, $BALANCE_t$ refers to the balance of an individual's loan at time $t$, while $BALANCE_{g,t}$ refers to the loan balance for an individual belonging to sensitive group $g$ at time $t$. Similarly, other features in the individual's vector, such as CURRINSTALL, DTI, or FICO scores, will be indexed by subscripts to refer to specific individuals or groups at different points in time.

Further details on how each agent affects these features are provided in the following discussion.

**Admissions Agent ($\boldsymbol{\pi}_1$):** At time step $t$, the Admissions Agent samples a group of $N_{1,t}$ applicants to form the application population for this time step and is tasked with deciding which of these applicants should be approved or rejected for a loan. Let $\mathbf{V} \in \mathcal{V}$ represent the matrix whose rows represent the feature vectors associated with these $N_{1,t}$ individuals. A scoring function $\mathbf{s} : \mathbb{R}^k \to [0, 1]$ produces a score which represents how qualified an individual is for repaying the loan that they have requested. The Admissions Agent, $\boldsymbol{\pi}_1 : \mathcal{V} \to [0, 1]^g$, is tasked with setting $g$ thresholds used to determine which individuals are admitted or rejected from the system. Two configurations of the agent's action space are considered: $g = 1$ ($g = 2$) indicates that the agent outputs a global (group-specific) threshold for approving individuals for loans at time step $t$. Admitted individuals are removed from the application population and enter the next phase of the loan system where they wait for their funds to be disbursed starting in time step $t + 1$. Rejected individuals are returned to the population and wait for another opportunity to be sampled and considered for a loan.

**Disbursement Agent ($\boldsymbol{\pi}_2$):** Once a person has been approved for a loan, he/she is removed from the application population pool and enters the funds disbursement stage of the pipeline. At time step $t$, $N_{2,t}$ individuals comprise the waiting population and wait in a queue for their funds to be disbursed. There is a fixed cap on the number of individuals who may have their funds disbursed at any given time step, which is used to mimic the real-world human resource constraints of a bank. Let $\mathbf{D} \in \mathcal{D}$ represent the matrix whose rows represent the feature vectors associated with these $N_{2,t}$ individuals. The Disbursement Agent, $\boldsymbol{\pi}_2 : \mathcal{D} \to [0, 1]^{N_{2,t}}$, reorders the queue at every time step by producing a score in the range $[0, 1]$ for every customer waiting for their funds to be disbursed. At each time step, the queue is re-sorted in descending order of the scores produced by this agent. Individuals at the top of the queue are then provided with funds until the disbursement cap is hit.

**Debt Management Agent ($\boldsymbol{\pi}_3$):** Once individuals receive their funds, they enter the loan repayment phase of the pipeline. At time step $t$, $N_{3,t}$ individuals in the repayment population make payments on their loans. Let $\mathbf{B} \in \mathcal{B}$ represent the matrix whose rows are the feature vectors associated with these $N_{3,t}$ individuals. Each individual is required to make payments according to a fixed payment schedule until their loan reaches maturity or they default. To support customers and reduce the likelihood of default, the Debt Management Agent, $\boldsymbol{\pi}_3 : \mathcal{B} \to [0, 1]^g$, can adjust repayment terms to alleviate financial strain. Two configurations of the agent's action space are considered: $g = 1$ ($g = 2$) indicates that the agent outputs a global (group-specific) adjustment percentage for the installments of all individuals repaying their loans at time step $t$. Once an individual's loan is terminated, they reenter the application population pool, from which the bank samples individuals for future loans.

**Reward and Disparity Component Indicators:** At the end of time step $t$, the environment returns a collection of reward and disparity component indicators used for reward and fairness violation measurement. A summary of these indicators is provided in Table 3. Each agent in this environment represents a functioning part of one institution, namely, a bank which has one primary objective—maximizing profits ($P_t$). Thus, the total amount of money made by the bank at time step $t$ represents the primary reward returned by the environment. Two other rewards can constructed from this list of indicators to guide learning models to

Table 3: `MAFE-Loan` Component Indicators

| Indicator | Description |
|---|---|
| $P_t$ | Bank profits at time step $t$ |
| $N_{L,t}^g$ | Number of people who applied for loans from Group $g$ at time step $t$ |
| $N_{A,t}^g$ | Number of people approved for loans from Group $g$ at time step $t$ |
| $N_{D,t}^g$ | Number of people from Group $g$ that had their fund disbursed at time step $t$ |
| $N_{T,t}^g$ | Sum of the number of time steps waited to receive loan funds for everyone from Group $g$ that received their funds at time step $t$. |
| $N_{R,t}^g$ | Number of terminated loans by members of Group $g$ at time step $t$. |
| $N_{F,t}^g$ | Number of defaulted loans by members of Group $g$ at time step $t$. |

avoid poor local minima; namely overall admissions rates ($\frac{\sum_t \sum_g N_{A,t}^g}{\sum_t \sum_g N_{L,t}^g}$) and (negative) overall default rates ($-\frac{\sum_t \sum_g N_{F,t}^g}{\sum_t \sum_g N_{R,t}^g}$).

The remaining environmental indicators provided by the system are used to measure fairness violations by tracking disparities among different rates provided for each demographic group at time step $t$. In particular, this information can be used to analyze three fairness disparities within the system among the two sensitive groups; namely, we can analyze disparities in: admissions rates ($\frac{\sum_t N_{A,t}^g}{\sum_t N_{L,t}^g}$), funds disbursement wait times ($\frac{\sum_t N_{T,t}^g}{\sum_t N_{D,t}^g}$), and default rates ($\frac{\sum_t N_{F,t}^g}{\sum_t N_{R,t}^g}$). Hence the indicators provided by the environment at each time step are used to measure three rewards and three fairness disparities.

**Mathematical Modeling:** A variety of environmental dynamics must be accounted for explicitly to ensure that the different underlying processes within the loan system function properly. These include modeling things such as a customer's financial rating or qualification to repay a loan, which is used by the Admissions Agent to set a threshold to determine who is and is not approved for a loan; loan payment schedule, which determines the amount a customer's loan installment at a given time step; and propensity to make a payment, which ultimately will determine whether or not he/she defaults. These design choices are outlined as follows.

Customer Qualification Scores:

A logistic regression is trained to take a customer's feature vector, $\mathbf{v}$, and produce a score in the range, $[0, 1]$. This model uses only a features from the Lending Club dataset, excluding any features from Table 3 augmented from environmental dynamics.

Payment Schedule:

Each loan is characterized by its duration (in time steps, representing its maturity), denoted as TIMETOMATURITY$_t$; interest rate, INTRATE; and initial balance, BALANCE$_{t_0}$. For simplicity, we respectively use $m$, $r$, and $B$ to refer to these variables in the ensuing discussion. At each time step, the customer is requested to make a payment, $Y_t$. In response, the customer will make a payment, $X_t$, where $0 \le X_t \le Y_t$. A payment below $Y_t$ indicates that the customer is falling behind on their loan obligations. The loan balance at each time step is updated using the following recursive formula:

$$B_t = (1 + r)B_{t-1} - X_t \tag{6}$$

The bank's goal is for the loan to be fully repaid by its maturity date, $m$. Assuming a fixed-rate payment schedule, at time step $t$, the payment request, $Y_t$, is set so that, if the customer were to pay the full amount of $Y_t$ at each time step until maturity, the loan balance would reach zero by time step $m$. To calculate this payment, we expand $B_m$ in terms of $B_t$

as follows:

$$B_m = (1+r)B_{m-1} - Y_t$$

$$= (1+r)^{m-t}B_t - \sum_{k=0}^{m-t-1} Y_t(1+r)^k$$

$$= (1+r)^{m-t}B_t - Y_t\frac{(1+r)^{m-t}-1}{r}$$

$$\tag{7}$$

Setting this equation equal to zero and solving for $Y_t$ yields the required payment amount, which depends on the loan's current balance, the interest rate, and the time remaining until maturity:

$$Y_t = \frac{r}{1-(1+r)^{t-m}}B_t$$

This payment ensures that, if paid in full at each time step, the loan balance will be entirely paid off by the maturity date, $m$.

Customer Payment:

The following equation is used to calculate the payment received by the bank on the installment requested at time step $t$:

$$X_t = clip(p_t + N_t, 0, 1) \cdot Y_t, \tag{8}$$

where $p_t$ is a propensity score that represents the percentage of $Y_t$ that a customer is willing to pay and $N_t \sim \mathcal{N}(\mu, \sigma^2)$ is Gaussian noise used to make the propensity score stochastic. The propensity scores for a customer are produced by a linear regression model trained to take the subset of a customer's feature vector, $\mathbf{v}$, containing features from the Lending Club dataset as input and output a percentage in the range $[0, 1]$. The labels for training this model are constructed by dividing the number of months it took for an individual's loan to terminate by the term of the loan for each individual in the training dataset. If the individual did not default, this label value is 1 (meaning they are completely likely to repay their loan). Moreover, the propensity scores of customers that default much earlier are lower than those of the customers that took a longer time to default.

Customer Default:

Default occurs if the applicant falls behind by more than 10% on all payments that the bank has requested from them for at least two consecutive time steps.

Bank Lending & Profits:

To finance the loans provided to its customers, we assume that the bank "borrows" money. That is, the bank pools deposits on which it, too, pays interest. Its profits are thus made by paying a lower interest rate than the rate it charges its customers. Thus the profits at a given time step are calculated as the difference between the sum of the payments received on the outstanding loans of its customers and the amount it is required to pay to its depositors.

**Loan Termination Feature Update Rule:** In reality, termination of a loan impacts an individual's financial well-being. For example, defaulting on a loan may reduce a person's FICO score, but the reverse may happen should a person repay his/her loan. Thus, each time a loan is terminated in this MAFE, we adjust a subset of features in $\mathbf{v}_v$ to reflect such a change, with the cause of termination (repayment versus default) determining whether the features will deteriorate or improve. In particular, we apply the following linear feature update rule to adjust these values:

$$\mathbf{v}_v = \begin{cases} \mathbf{v}_v + \mathbf{c} & \text{, if Customer Repays Loan} \\ \mathbf{v}_v - \mathbf{c} & \text{, if Customer Defaults on Loan} \end{cases} \tag{9}$$

for some constant vector $c$.

**Episode Termination:** An episode in MAFE-Loan may terminate for two reasons: (1) The maximal number of time steps set by a user has been reached and (2) the bank goes bankrupt. Bankruptcy occurs if at any point during the simulation, the total amount of money lost by the bank is greater than the total amount of money it has received.

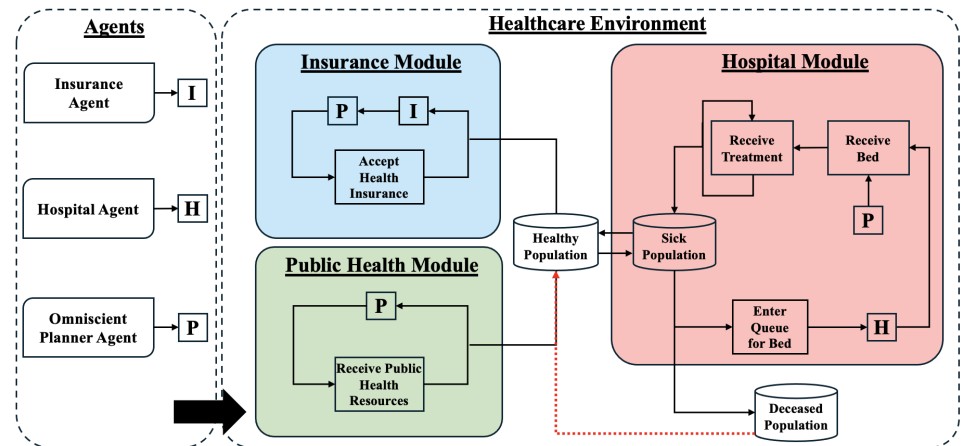

Figure 13: `MAFE-Health` Diagram

## G `MAFE-HEALTH` MODELING DETAILS

In this section we provide a detailed explanation of how we design `MAFE-Health` introduced in Section 4 of the main paper.

**Overview:** A diagram illustrating the design of `MAFE-Health` is shown in Figure 13. This environment models the interactions among three core agents: an insurance company, a hospital, and a central planner. These agents collectively impact the health and insurance coverage of the population.

At each time step, the Insurance Agent offers a premium to each individual, who decides whether to accept the plan based on its cost. The premium affects the likelihood of obtaining insurance, which influences the individual's access to routine medical care. Thus, uninsured individuals face greater health risks due to limited access to early disease detection and regular treatment.

Individuals are categorized into three health states: **healthy**, **ill**, and **deceased**. Healthy individuals may become ill, and sick individuals may either recover or pass away. Upon diagnosis, a sick individual joins a hospital queue, where they await treatment. The allocation of hospital beds depends on the hospital's capacity, with individuals prioritized for treatment according to the queue-ordering scores produced by the Hospital Agent. The likelihood of recovery is higher for individuals who are treated early, which is more likely if they are insured.

The Central Planner Agent allocates a healthcare budget at each time step, distributing funds across hospital infrastructure, public health initiatives, and insurance subsidies. The planner may also save funds for future investments in the healthcare system.

When moralities occur, deceased individuals are reintroduced into the population to simulate real-world population replenishment. However, in contrast with the Loan MAFE, where all agents act at every time step, in this system, the Hospital Agent acts at every time step, while the Insurance and Central Planner Agents take actions every $k$ time steps. This reflects real-world scenarios where premiums and budgets are set periodically, while healthcare needs can arise at any time.

Ultimately, the collective decisions made by these agents affect mortality rates within the system. In the following sections, we provide a detailed description of the roles and operations of each agent within the environment at a given time step $t$.

**Population:** At the beginning of the healthcare simulation, a global population is initialized which consists of $N$ healthy individuals, each of whom has an associated global feature vector $\mathbf{v} = [\mathbf{v}_c^T \quad \mathbf{v}_v^T]^T \in \mathbb{R}^k$ which contain **all** demographic information and indicators correlated with a person's health which the agents use to make their decisions. $\mathbf{v}_c$ represents the subset

Table 4: `MAFE-Health` Features

| Variable | Origin | How it is updated | Description |
|---|---|---|---|
| YEAR | IPUMS MEPS | None | Survey Year |
| AGE | IPUMS MEPS | None | Age |
| SEX | IPUMS MEPS | None | Sex |
| REGION | IPUMS MEPS | None | Census region as of 12/31 of the survey year |
| FAMSIZE | IPUMS MEPS | None | Number of persons in family |
| RACE | IPUMS MEPS | None | Main racial background |
| USBORN | IPUMS MEPS | None | Born in United States |
| EDUC | IPUMS MEPS | None | Educational Attainment |
| HICOV | IPUMS MEPS | Insurance Agent ($\pi_1$) | Has health insurance |
| CHOLHIGHEV | IPUMS MEPS | None | Ever told had high cholesterol |
| SMOKENOW | IPUMS MEPS | None | Smoke cigarettes now |
| INCTOT | IPUMS MEPS | Central Planner Agent ($\pi_3$) | Total personal income |
| FTOTVAL | IPUMS MEPS | Central Planner Agent ($\pi_3$) | Total family income |
| POVLEV | IPUMS MEPS | Central Planner Agent ($\pi_3$) | Family income as a percentage of the poverty line |
| AEFFORT | IPUMS MEPS | Central Planner Agent ($\pi_3$) | Felt everything an effort, past 30 days |
| ANERVOUS | IPUMS MEPS | Central Planner Agent ($\pi_3$) | How often felt nervous, past 30 days |
| ARESTLESS | IPUMS MEPS | Central Planner Agent ($\pi_3$) | How often felt restless, past 30 days |
| AHOPELESS | IPUMS MEPS | Central Planner ($\pi_3$) | How often felt hopeless, past 30 days |
| ASAD | IPUMS MEPS | Central Planner ($\pi_3$) | How often felt sad, past 30 days |
| AWORTHLESS | IPUMS MEPS | Central Planner Agent ($\pi_3$) | How often felt worthless, past 30 days |
| HEALTH | IPUMS MEPS | Environment Dynamics | Health status |
| NEEDBED | Environment | Environment Dynamics | Waiting for hospital bed |
| INHOSP | Environment | Hospital Agent ($\pi_2$) | Person is in the hospital |
| ILLNESS | Environment | Environment Dynamics | How long person has been ill |
| DECEASED | Environment | Environment Dynamics | Person is deceased |
| NGEOBED | Environment | Environment Dynamics | Number of beds in each region |
| HIPCOST | Environment | Environment Dynamics | Health insurance premium |
| HIPFULLCOST | Environment | Environment Dynamics | Amount paid to health insurance by all members in same region |
| HOSPCOST | Environment | Environment Dynamics | Cost of hospital stay |
| WAITBED | Environment | Environment Dynamics | Waiting for a bed |
| ILLTIME | Environment | Environment Dynamics | How long sick with current illness |
| PLANBUDGET | Environment | Environment Dynamics | Central Planner current budget |

of constant features in $\mathbf{v}$ which remain constant throughout the entire simulation, while $\mathbf{v}_v$

represents a person's variable features which are updated based on the actions made by the different agents.

To ensure that data we use contain realistic features, we use realworld census data curated from the Integrated Public Use Microdata Series (IPUMS) Medical Expenditure Panel Survey (MEPS) available under IPUMS Health Surveys Blewett et al. (2024). Our population is constructed from survey responses from 2014 to 2016. These responses are converted to feature vectors using the variables listed in Table 4. All responses that contain missing values for any survey questions associated with these variables are filtered from the population. Each of these feature vectors is then augmented to include information associated with the dynamics of the environment, such as INSURED, which specifies whether or not a person has insurance at a particular time step.

The variables in $\mathbf{v}_v$ may be influenced by the actions taken by different agents. For example, public health subsidies funded by the Central Planner Agent can improve general health variables, while insurance subsidies can increase the likelihood of an individual having health coverage. These evolving features provide the necessary observations for the agents to adjust their strategies at each time step.

In the remainder of this section, we use subscript notation to refer to the value of a particular variable for an arbitrary individual or group at time $t$. For instance, $\text{INCTOT}_t$ refers to the total income of an individual at time $t$, while $\text{INCTOT}_{g,t}$ refers to the total income of an individual in sensitive group $g$ at time $t$. Similarly, other variables such as insurance status (INSURED), health indicators, and demographic factors will be indexed with subscripts to track changes over time for specific individuals or groups.

Further details on how each agent influences these features are provided in the following discussion.

**Insurance Agent ($\boldsymbol{\pi}_1$):** Every $k$ time steps the Insurance Agent must decide to offer an insurance package containing of a set premium to all individuals in the global population. Let $\mathbf{V} \in \mathcal{V}$ represent the matrix whose rows represent the feature vectors associated with these $N$ individuals. The Insurance Agent, $\boldsymbol{\pi}_1 : \mathcal{V} \to [0,1]^N$, is responsible for determining the premium offered to each individual in the system by producing a value in the range $[0,1]$. This value is then scaled to establish a recurring premium over the next $k$ time steps, with the scaling factor ensuring that the premium falls within the allowable range, from 0 to the maximum permissible amount. Each customer then decides whether or not he/she will accept this premium for the duration of the ensuing cycle or not. We elaborate on how we model customer decisions in the mathematical modeling discussion we provide later in this section.

**Hospital Agent ($\boldsymbol{\pi}_2$):** Once a person becomes sick, they are reclassified from the healthy population to become part of the sick population. At time step $t$, $N_{2,t}$ individuals are waiting for a hospital bed. Let $\mathbf{D} \in \mathcal{D}$ represent the matrix whose rows represent the feature vectors associated with these $N_{2,t}$ individuals. The Hospital Agent, $\boldsymbol{\pi}_2 : \mathcal{D} \to [0,1]^{N_{2,t}}$, produces a score for each one of these individuals in the range $[0,1]$ which are used to reorder the global hospital queue (in descending order). The queue for each local hospital is then determined by segmenting the sorted scores of the individuals in the global hospital queue that belong to a particular geographic regions. Individuals with scores at the top of the queue are then provided with beds based on their local hospital's availability.

**Central Planner Agent ($\boldsymbol{\pi}_3$):** The Central Planner Agent makes decisions that improve outcomes for the different entities within the system by allocating its budget to three types of subsidies—insurance subsidies for customers, public health subsidies, and hospital infrastructure subsidies. To make informed decisions, it receives the feature information of the global population. Namely, let $\mathbf{D} \in \mathcal{D}$ represent the matrix whose rows represent the feature vectors associated with all $N_{3,t}$ individuals in the global population at time $t$ and assume that there are $N_g$ geographic regions in the environment. Then, the Central Planner Agent, $\boldsymbol{\pi}_3 : \mathcal{D} \to [0,1]^{3N_g+3}$, produces actions that can be represented by a tree structure, as illustrated in Figure 14. Given the Central Planner Agent's budget at time $t$, the first four elements of its action vector correspond with the middle level of nodes in this tree and represent the percentage of budget allocated to each of the three categories of subsidies and

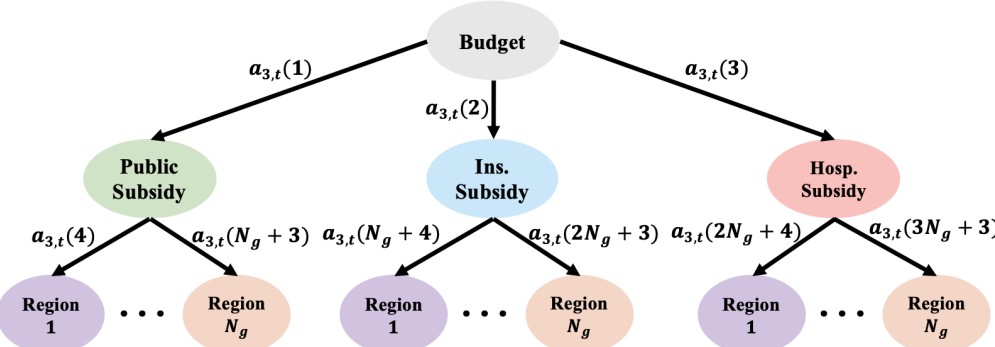

Figure 14: Action structure of Central Planner.

Table 5: `MAFE-Health` Component Indicators

| Indicator | Description |
|---|---|
| $P_t$ | Insurance profits at time step $t$ |
| $N_{g,t}^G$ | Total number of people in Region $g$ at time step $t$ |
| $N_{g,t}^I$ | Number of people insured in Region $g$ at time step $t$ |
| $N_{g,t}^H$ | Number of healthy people in Region $g$ at the start of time step $t$ |
| $N_{g,t}^S$ | Number of people who become sick in Region $g$ at time step $t$ |
| $N_{g,t}^T$ | Number of people whose illnesses terminated in Region $g$ at time step $t$ |
| $N_{g,t}^M$ | Number of moralities in Region $g$ at time step $t$ |

rollover funds for the next time step. The remaining $3N_g$ values represent the leaves of this tree and determine the percentage of each subsidy allocated to each of the $N_g$ geographic regions. Letting $a_{3,t}$ represent the action taken by the Central Planner Agent, $\boldsymbol{\pi}_3$, at time $t$, we have that $\sum_{i=0}^{3} a_{3,t}(i)$, $\sum_{i=4}^{N_g+3} a_{3,t}(i)$, $\sum_{i=N_g+4}^{2N_g+3} a_{3,t}(i)$, and $\sum_{i=2N_g+4}^{3N_g+3} a_{3,t}(i)$ should all equal 1. Thus, the product of actions taken along a path from the root of the tree to an arbitrary leaf provides the percentage of the agent's budget allocated to a particular subsidy in a given geographic region or rollover investment.

**Indicators for Measuring Rewards and Fairness:** At the end of time step $t$, the environment returns a collection of indicators used to measure rewards and fairness violations within the system. A summary of these indicators is provided in Table 5. These indicators can be used to construct the following set of rewards that motivate these agents in the real world: insurance profits ($P_t$), insured rates ($\frac{\sum_t \sum_g N_{g,t}^I}{\sum_t \sum_g N_{g,t}^G}$), (negative) incidence rates ($-\frac{\sum_t \sum_g N_{g,t}^S}{\sum_t \sum_g N_{g,t}^H}$), and (negative) mortality rates ($-\frac{\sum_t \sum_g N_{g,t}^M}{\sum_t \sum_g N_{g,t}^T}$).

The remaining environmental indicators provided by the system are used to measure fairness by tracking disparities in different rates over different geographic regions in the environment over time. In particular, this information can be used to analyze three fairness disparities within the system among $N_g$ geographic regions; namely, we can analyze disparities in insured rates ($\frac{\sum_t N_{g,t}^I}{\sum_t N_{g,t}^G}$), incidence rates ($\frac{\sum_t N_{g,t}^S}{\sum_t N_{g,t}^H}$), and mortality rates ($\frac{\sum_t N_{g,t}^M}{\sum_t N_{g,t}^T}$) across geographic regions using the standard deviation measure from Equation 4. Hence, the indicators provided by the environment at each time step are used to measure four rewards and three fairness disparities.

**Mathematical Modeling:**

Health Risk Scores:

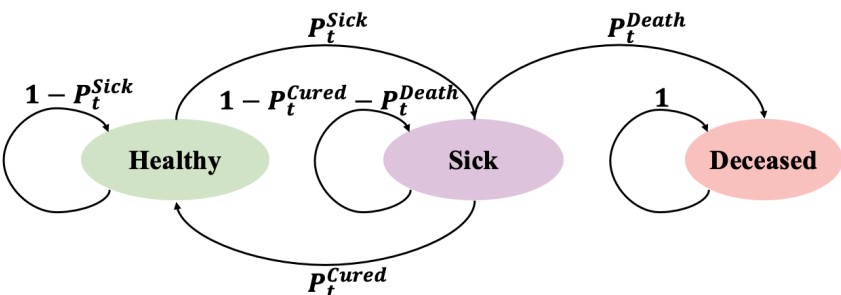

Figure 15: Health state transition.

A linear regression is trained to take a customer's feature vector at time $t$, $\mathbf{v}_t$, and produce a health risk score, $\text{HEALTH}_t$, in the range $[1, 5]$ using the IPUMs health dataset. A higher value of $\text{HEALTH}_t$ indicates that a participant has worse health and is thus at increased risk of illness at time $t$. To ensure that the outputs of the linear regression are bounded within this range, the final health score is given after applying the clip operation to the original health score outputs, e.g. $clip(\text{HEALTH}_t, 1, 5)$.

Health Transition Likelihoods:

An individual in this MAFE may transition across three health states in this simulation—namely, they may be healthy, ill, or deceased, as illustrated by the graph shown in Figure 15. At the beginning of the simulation, every individual resides in the healthy state. As an episode progresses, each person may transition between states according to the state transition probabilities. As depicted in Figure 15, let $P_t^{Sick}, P_t^{Death}$, and $P_t^{Cured}$ represent the conditional probabilities that individuals who are healthy become ill, individuals who are ill to pass away, and individuals who are ill become healthy at time $t$.

These transition probabilities are directly and indirectly influenced by the actions taken by the agents within the system. We model the likelihood of an individual who is not sick becomes sick as being positively correlated with a person having poor health (e.g. positively correlated with the value of $\text{HEALTH}_t$) and negatively correlated with having health insurance (e.g. negatively correlated with the binary value of $\text{HICOV}_t$, with a value of 1 indicating that a person has health insurance), given by the following equation:

$$P_t^{Sick} = A(1 - \text{HICOV}_t) + \frac{B}{5}\text{HEALTH}_t. \tag{10}$$

To ensure that $P_t^{Sick}$ is a probability, $A$ and $B$ must be chosen to ensure that $A + \frac{B}{5} \in [0, 1]$ (where the factor of 5 is included since $\text{HEALTH}_t \in [1, 5]$).

We model the probability that a sick person passes away, $P_t^{Death}$, as the product of two probabilities: the probability that their illness terminates, $P^{Terminate}$, and the probability that the termination is due to mortality (rather than recovery), $P^{Mortality}$. That is,

$$P_t^{Death} = P_t^{Terminate} P_t^{Mortality}. \tag{11}$$

Similarly, the probability that a person that is sick is cured is given by

$$P_t^{Cured} = P_t^{Terminate}(1 - P_t^{Mortality}). \tag{12}$$

Both $P_t^{Terminate}$ and $P_t^{Mortality}$ are modeled using an exponential family of functions of the form:

$$C + D^{E \cdot \text{ILLTIME}_t + F \cdot \text{WAITBED}_t + G \cdot \text{HEALTH}_t + H}, \tag{13}$$

where $\text{ILLTIME}_t$ represents the number of consecutive time steps that a person with an illness has had it as of time step $t$, $\text{WAITBED}_t$ represents the amount of time that a person who is ill had to wait before receiving a hospital bed as of time step $t$, and $\text{HEALTH}_t$ specifies a person's general health quality as of time step $t$.

We now provide the intuition we consider for making our parameter selections, though we note that this is only one way of modeling these probabilities. These parameter choices, and the functional forms, themselves, can be adapted by users of our MAFEs as they see fit.

We select $\text{ILLTIME}_t$ to be negatively correlated with $P_t^{Terminate}$ and positively correlated with $P_t^{Mortality}$ as an illness may be more likely to be resolved the longer one has it, but a longer illness could indicated it is more serve and may increase the likelihood that someone dies from it. One the other hand, and increase value of $\text{HEALTH}_t$ means someone has poorer overall health. Since it may take someone with poorer health more time to fend off an illness, putting them at increased risk of mortality, $\text{HEALTH}_t$ we specify its coefficient parameter to make it positively correlated with $P_t^{Terminate}$ and $P_t^{Mortality}$. Similarly, the longer it takes someone to receive a hospital bed, the longer and illness may fester since he/she may be unable to receive the appropriate care needed to cure it. As a result, we ensure that $\text{WAITBED}_t$ is positively correlated with $P_t^{Terminate}$ and $P_t^{Mortality}$.

Cost of Hospital Infrastructure:

Hospital infrastructure refers to the physical facilities needed to increase the number of available beds in a hospital. Building new infrastructure involves two main costs: a base cost, which is incurred for any construction plan, and a proportional cost, which depends on the number of new beds being built. The total cost of building new infrastructure is modeled as a linear function, where the base cost is added to the cost that increases with the number of new beds. This creates a trade-off for the Central Planner Agent, which must decide when to invest in infrastructure. Investing in small projects repeatedly can become expensive due to the base cost, while waiting to fund a larger project may lead to insufficient hospital resources and more deaths.

Time to Build Hospital Infrastructure:

The time required to build new hospital infrastructure is modeled similarly to the cost of infrastructure, with a different interpretation of the variables. The time required for construction depends on the size of the project. There is a base amount of time required for planning and setting up the project, and additional time required is linearly proportional to the number of new beds added by the project.

Individual's Likelihood of Accepting Insurance:

An individual's willingness to pay for insurance depends on a number of factors whether or not his/her insurance premiums is reasonably priced (which is relatively determined by a person's financial well-being, e.g. their net worth), their age, and their health, the size of their family, and so on. To strike a balance between complexity and fidelity, we model this as a function of the following factors: net family income ($\text{FTOTVAL}_t$), household size ($\text{FAMSIZE}_t$), and the monthly premium ($\text{HIPCOST}_t$) a customer would be required to pay should he/she accept health insurance. This is done by sampling a Bernoulli distribution, $\text{Bernoulli}(P_t^{Insured})$, where $P_t^{Insured}$ is given by:

$$P_t^{Insured} = 1 - e^{\frac{\text{FTOTVAL}_t}{\text{HIPCOST}_t(\text{FAMSIZE}_t)}}. \tag{14}$$

Distributing Insurance Subsidies:

The final premium for health insurance that a customer is offered is determined by subtracting the amount subsidized by the Central Planner Agent from the initial price set by the Insurance Agent. However, rather than making case-by-case decisions on subsidy allocation, the Central Planner Agent designates a fixed budget for subsidizing insurance within each geographic region, as described in the description of the Central Planner Agent. A rule is then applied to distribute these funds proportionally to all individuals within each region. Specifically, subsidies are inversely weighted by each individual's per capita household income. Let $\text{FTOTVAL}_{g,t}(i)$ represent the per capita income of the $i^{th}$ individual among $N_g$ members living in Region $g$ at time $t$. The fraction of the total subsidy allocated to this individual is calculated as:

$$w_i = \frac{\frac{1}{\text{FTOTVAL}_{g,t}(i)}}{\sum_{n=1}^{N_g} \frac{1}{\text{FTOTVAL}_{g,t}(n)}}. \tag{15}$$

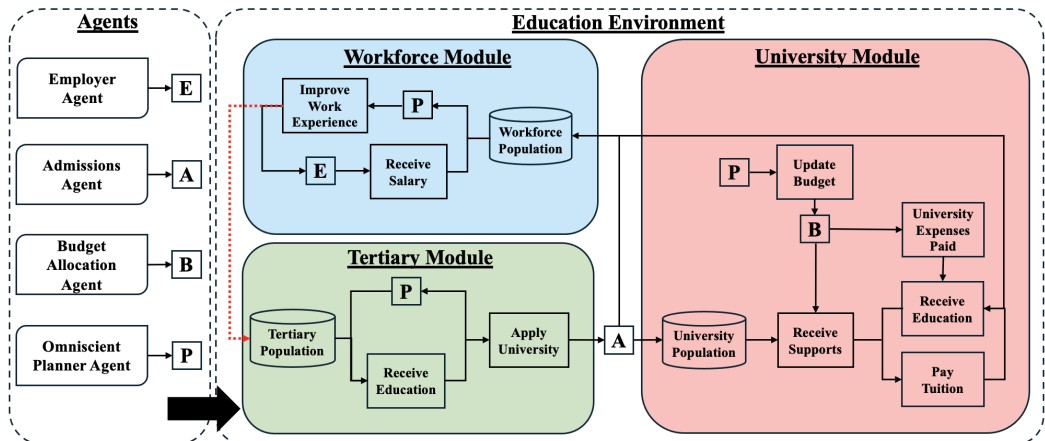

Figure 16: `MAFE-Edu` Diagram

Effect of Public Health Investment:

In each time step, a subset of the updateable features in $\mathbf{v}_v$ associated with each individual in Region $g$ will improve with probability $P_{g,t}^{improve}$, remain unchanged with constant probability $U$, or deteriorate with probability $1 - P_{g,t}^{improve} - U$. We treat $U$ as a user specified constant. The value of $P_{g,t}^{improve}$ is affected by the amount of the Central Planner Agent's budget that is used on public health expenditures in Region $g$ at time step $t$. In particular, we model $P_{g,t}^{improve}$ as a function of the amount of the planners budget invested in the region in which this individual is located at time $t$. For constant hyperparameters $Q, R, V,$ and $W$, this is given by the following equation:

$$P_{g,t}^{improve}(x) = Q + R\sigma(V \cdot x + W) \tag{16}$$

where $\sigma$ represents a sigmoid function. We assume this equation is tuned so that $P_{g,t}^{improve}$ is non-negative and

$$\sup_x P_{g,t}^{improve}(x) + U = 1. \tag{17}$$

To determine if an individuals features improve, deteriorate, or remain unchanged we sample a uniform distribution over the range $[0, 1]$ and update the features appropriately based on the segment in which the output value lands—$[0, P_{g,t}^{improve}]$, $(P_{g,t}^{improve}, P_{g,t}^{improve} + U]$, or $(P_{g,t}^{improve} + U, 1]$.

**Episode Termination:** An episode may terminate for three reasons. First, if the agents produce actions that lead them to successfully reach the user specified terminal time step, the episode terminates. Conversely, the environment may also terminate early if any entity in the institution fails. Particularly, if the Insurance Agent ever has net negative profits. at is, if the income it receives from premium payments is outweighed by the cost of paying for customer's hospital stays over the entirety of an episode. The episode also fails if the entire living population in the simulation is depleted, we consider the episode a failure.

## H   MAFE-Edu MODELING DETAILS

**Overview:** A diagram outlining the design of `MAFE-Edu` is provided in Figure 16. This environment is designed to simulate the school-to-employment pipeline by modeling three key entities involved in this process: a university system, the employers of each individual, and an central planner (which functions as a central planner or government-like entity). The Central Planner Agent ($\boldsymbol{\pi}_4$) and Employer Agent ($\boldsymbol{\pi}_3$) are both modeled using a single agent. However, two separate agents are used to model distinct processes within the university: the Admissions Agent ($\boldsymbol{\pi}_1$), which determines which applicants are admitted or rejected, and

the University Budget Allocation Agent ($\boldsymbol{\pi}_2$), which decides how to allocate the university's budget across various expenses. The decisions made by these agents collectively shape the students' future success.

At each time step, the population is categorized into three groups: the **tertiary population** (individuals not actively involved in the simulation), the **higher education population** (degree-seeking students within the university system), and the **working population** (individuals employed in the workforce). The tertiary population consists of individuals who are not currently involved in the higher education pipeline. At each time step, a subset of these individuals is sampled from the tertiary population, with each passing through the education system for a fixed number of time steps, representing their journey from enrollment to career termination, before being returned to the tertiary population for future resampling.

When individuals sampled from the tertiary population apply to college, the University Admissions Agent decides who will be accepted into the higher education system to pursue one or more degrees. Those who are rejected immediately enter the workforce. At any given time step, an individual within the university system may choose to exit and join the workforce, with the length of time they have spent in the university system determining the highest degree they have earned. The longer they stay in the university, the higher the degree attained.

The number of individuals the university can accept and support successfully depends on the University Budget Allocation Agent, which determines how the university allocates the funds it has accrued at each time step. These funds are distributed across various resources that the university believes will lead to the best student outcomes, as measured by the rewards provided by the system.

The Central Planner Agent also operates with a budget at each time step, which it allocates across various expenditures that influence individuals' educational and career success. These expenditures include tertiary investments (which improve the quality of education children receive in their formative years), university budget investments (which serve as a secondary source of funding, aside from tuition), and diversity incentives (which may be provided to the employer agent to encourage salary equity in the workforce).

Once an individual enters the workforce, they remain there until the number of time steps they have spent in the simulation reaches the limit, $N$. During this time, the Employer Agent sets the salary for each worker, which directly affects their productivity. Upon reaching the terminal time step, the individual is removed from the environment, their features are updated, and they are returned to the tertiary population, where they may be resampled for a future pass through the system. This process continues until the episode is terminated.

Ultimately, the collective decisions made by these agents determine individuals' academic and career success within the system. In the following sections, we provide a detailed description of the roles and operations of each agent at a given time step, $t$.

**Population:** At the beginning of the education simulation, a global population is initialized which consists of $N$ individuals, each of whom has an associated global feature vectors, $\mathbf{v} = [\mathbf{v}_c^T \quad \mathbf{v}_v^T]^T \in \mathbb{R}^k$ which contain **all** demographic information and indicators correlated with a person's experience and academic merits which the agents use to make their decisions. $\mathbf{v}_c$ represents the subset of constant features in $\mathbf{v}$ which remain constant throughout the entire simulation, while $\mathbf{v}_v$ represents a person's variable features which are updated based on the actions made by the different agents.

To ensure that data we use contain realistic features, we use real-world census data curated from the Integrated Public Use Microdata Series (IPUMS) Higher Ed (EDUC) Surveys Minnesota Population Center (2016). Our population is constructed from survey responses from 2014 to 2016. These responses are converted to feature vectors using the variables listed in Table 6. All responses that contain missing values for any survey questions associated with these variables are filtered from the population. Each of these feature vectors is then augmented to include information associated with the dynamics of the environment, such as TIMEINUNIV, which specifies the amount of time an individual has spent in the university through the current time step.

The variables in $\mathbf{v}_v$ may be influenced by the actions taken by different agents. For example, if the university detects structural performance disparities among different demographic groups, it could allocate more of its budget to providing mentorship programs to the disadvantaged group, thereby increasing their likelihood of obtaining higher GPAs and affecting the CURRENTGPA feature. Alternatively, the Central Planner could allocate funds for employer incentives to mitigate salary-based disparities among members of different demographic groups, thus affecting the SALARY feature.

In the remainder of this section, we use subscript notation to refer to the value of a particular variable for an arbitrary individual or group at time $t$. For example, $\text{GPA}_t$ refers to an individual's cumulative GPA at time $t$, while $\text{GPA}_{g,t}$ refers to the GPA of an individual with sensitive attribute $g$ at time $t$. This subscript notation allows us to track how variables, such as GPA and time in university, evolve over time for specific individuals or groups, including those based on demographic characteristics.

Further details on how each agent influences these features are provided in the following discussion.

**University Admissions Agent ($\boldsymbol{\pi}_1$):** Different from the standard ML setup in which an admissions agent is represented by a classifier who accepts any students whose scores fall above a given (typically 0.5) threshold, we take a resource constrained approach to modeling admissions. In particular, we assume that for the university to provide quality instruction to students, there is a cap on the size of the student-instructor ratio. Thus, there is a limit to the number of students that may be admitted to the university at time $t$ which depends on the number of students already in the university and the number of instructors employed by the university at time $t$. At the same time, it is essential for the university to raise money to pay for expenses such as teacher salaries and infrastructure. Thus, the university should always admit as many students as it can without violating the student-instructor ratio cap so as to ensure that no available classroom seats are left empty. With this in mind, our admission agent operates as follows.

At time step $t$, a collection of $N_{1,t}$ individuals are sampled from the tertiary population to apply for college. Let $\mathbf{D} \in \mathcal{D}$ represent the matrix whose rows represent the feature vectors associated with these $N_{1,t}$ individuals. The admissions agent, $\boldsymbol{\pi}_1 : \mathcal{D} \rightarrow [0,1]^{N_{1,t}}$, produces a score for each of these individuals in the range $[0,1]$, which is used to rank students in terms of who the university most desires to admit. Students are then admitted in order of their rank until all available slots at the university have been filled. Those who are rejected immediately enter the workforce.

**University Budget Allocation Agent ($\boldsymbol{\pi}_2$):** The University Budget Allocation Agent makes decisions that affect the proper functioning of the university, which have consequences for student success. In particular, given a budget, this agent allocates these funds to four primary expenses—university infrastructure, staff salaries, scholarships, and minority mentorship programs which have the potential to improve the performance of underrepresented groups in higher education. To make informed decisions, it receives the feature information for the higher education population. Namely, let $\mathbf{D} \in \mathcal{D}$ represent the matrix whose rows represent the feature vectors associated with all $N_{2,t}$ students currently in the university system at time $t$. Then, the University Budget Allocation Agent, $\boldsymbol{\pi}_2 : \mathcal{D} \rightarrow [0,1]^5$, produces four actions that represent the percentages of its budget that are allocated to each of the four expenses it is allowed to pay, plus an amount that it is allowed to roll over for future budgeting, such as for investing in larger infrastructure projects than it currently can afford. Thus, letting $a_{2,t}$ represents the actions taken by the University Budget Allocation Agent, $\boldsymbol{\pi}_2$, at time $t$, we have that this action must be constrained such that $\sum_{i=1}^{5} a_{2,t}(i)$ equals 1.

**Employer Agent ($\boldsymbol{\pi}_3$):** At time step $t$, the workforce population consists of $N_{3,t}$ people, for each of whom the Employer Agent provides a salary. Let $\mathbf{V} \in \mathcal{V}$ represent the matrix whose rows represent the feature vectors associated with these $N_{3,t}$ individuals. The Employer Agent, $\boldsymbol{\pi}_3 : \mathcal{V} \rightarrow [0,1]^{N_{3,t}}$, is responsible for determining the salary for each individual in the workforce by producing a value in the range $[0,1]$. This value is then scaled to establish an annual salary for the ensuring time step, with the scaling factor ensuring that the salary falls within the allowable range, from 0 to the maximum permissible amount. Here, the employer

Table 6: `MAFE-Edu` Features

| Variable | Origin | How it is Updated | Description |
|---|---|---|---|
| SEX | IPUMS EDUC | None | Sex |
| MINRTY | IPUMS EDUC | None | Minority indicator |
| RACE | IPUMS EDUC | None | Main racial background |
| NBAMEMG | IPUMS EDUC | None | Field of major first degree |
| NDGMEMG | IPUMS EDUC | None | Field of major highest degree |
| REGION | IPUMS EDUC | None | Region of the country lived in |
| NOCPRMG | IPUMS EDUC | None | Job code for principal job (major group) |
| SALARY | IPUMS EDUC | Employer ($\pi_3$) | Salary (annualized) |
| HRSWK | IPUMS EDUC | Central Planner ($\pi_4$) | Principal job hours worked |
| EMSEC | IPUMS EDUC | Central Planner ($\pi_4$) | Employer sector |
| EMSIZE | IPUMS EDUC | Central Planner ($\pi_4$) | Size of employer |
| UGLOAN | IPUMS EDUC | Central Planner ($\pi_4$) | Total amount taken out for undergraduate loans |
| GRLOAN | IPUMS EDUC | Central Planner ($\pi_4$) | Total amount taken out for graduate loans |
| DGRDG | IPUMS EDUC | Environment Dynamics | Type of highest certificate or degree |
| GPA | IPUMS EDUC | Environment Dynamics, Central Planner ($\pi_4$) | Cumulative College GPA |
| INENV | Environment | Environment Dynamics | Indicator specifying if person was sampled to become part of the environment |
| INWORKF | Environment | Environment Dynamics | Indicator specifying if person in environment is in workforce |
| INUNIV | Environment | Environment Dynamics | Indicator specifying if person in environment is in university |
| INMINTYPGRM | Environment | Environment Dynamics | Indicator specifying if person in university if in minority mentorship program |
| CURRENTGPA | Environment | Environment Dynamics | GPA of student in university at current time step |
| PLANBUDGET | Environment | Environment Dynamics | Central planner current budget |
| UNIVBUDGET | Environment | Environment Dynamics | University's current budget |
| ANNUALTUIT | Environment | Environment Dynamics | Student's annual tuition (scholarship adjusted) |
| N_UNIV_UNITS | Environment | Environment Dynamics | Number of university infrastructure units |
| N_FACULTY | Environment | Environment Dynamics | Number of university faculty |
| N_STUDENTS_CURR | Environment | Environment Dynamics | number of students in university |
| TIMEINUNIV | Environment | Environment Dynamics | Time student has spent in university (nonzero if INENV=1 and INUNIV=1) |
| TIMEINWORKF | Environment | Environment Dynamics | Number of time steps person has been in university (nonzero if INENV=1 and INWORF=1) |
| TIMEINENV | Environment | Environment Dynamics | Number of time steps person has been in environment (nonzero if INENV=1) |
| DIVINVEST | Environment | Environment Dynamics | Amount of money Central Planner allocates to employer diversity incentives |
| AGE | Environment | Environment Dynamics | Age of person in environment |
| AVE_SALARY | Environment | Environment Dynamics | Average salary of person over entirety of work career |

agent is not meant to be interpreted as a single employer. Rather, it can be thought of as a tool that decides the salary of a particular person for the job at which they work, whatever

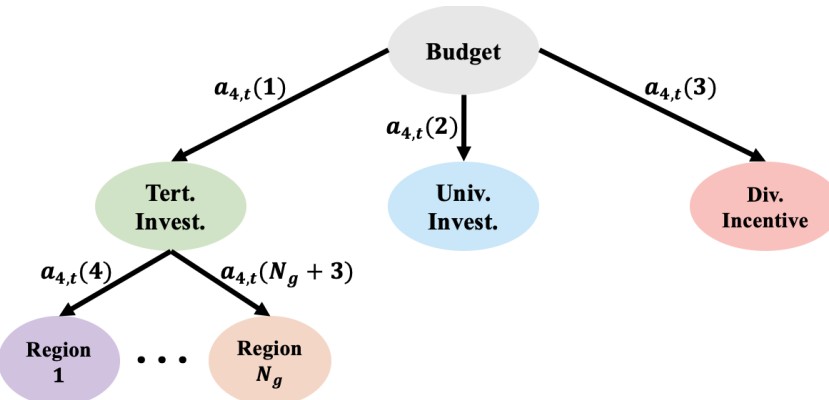

Figure 17: Action structure of Education Central Planner.

that job may be. The goal of this agent is to set this salary so that the utility the employer received from each worker is maximized. We elaborate on how we quantify utility in our ensuing discussion.

**Central Planner Agent ($\pi_4$):** The Central Planner Agent makes decisions that improve outcomes for the different entities within the system by allocating its budget for three types of investments—investments in tertiary education resources, university funding, and diversity incentives for employers. To make informed decisions, it receives the feature information of the global population. Namely, let $\mathbf{D} \in \mathcal{D}$ represent the matrix whose rows represent the feature vectors associated with all $N$ individuals in the global population at time $t$ and assume that there are $N_g$ geographic regions in which these students may have received their tertiary education in the environment. Then, the Central Planner Agent, $\pi_4 : \mathcal{D} \to [0,1]^{N_g+3}$ produces actions that can be represented by a tree structure, as illustrated in Figure 17. Given its budget at time $t$, $B_t$, the first three elements of its action vector correspond with the middle level of nodes in this tree and represent the percentage of $B_t$ allocated to each of the three investment categories. Note, no rollover action is provided to this agent since there are no incentives for it to budget for future investment. The remaining $N_g$ values represent the leaves under the tertiary investment node in Figure 17 and determine the percentage of tertiary investment allocated to each of the $N_g$ geographic regions. Letting $a_{4,t}$ represent the action taken by the Central Planner Agent at time $t$, we have that $\sum_{i=1}^{3} a_{4,t}(i)$ and $\sum_{i=4}^{N_g+3} a_{4,t}(i)$ should all equal 1.

**Indicators for Measuring Rewards and Fairness:**

At the end of time step $t$, the environment returns a collection of indicators used to measure rewards and fairness violations within the system. A summary of these indicators is provided in Table 7. These indicators can be used to construct the following set of rewards that motivate these agents in the real world: employer profits ($P_t$), admissions rates ($\frac{\sum_t \sum_g E_{g,t}^U}{\sum_t \sum_g A_{U,t}^g}$), and graduation rates for undergraduate, Master's, or doctoral degrees ($\frac{\sum_t \sum_g G_{g,t}^U}{\sum_t \sum_g C_{g,t}^U}$, $\frac{\sum_t \sum_g G_{g,t}^M}{\sum_t \sum_g C_{g,t}^M}$, and $\frac{\sum_t \sum_g G_{g,t}^D}{\sum_t \sum_g C_{g,t}^D}$), and average salaries ($\frac{\sum_t \sum_g S_{g,t}}{\sum_t \sum_g W_{g,t}}$).

The remaining environmental indicators provided by the system are used to measure fairness by tracking disparities among different rates provided for each demographic group at time step $t$. In particular, this information can be used to analyze five fairness disparities within the system among the two sensitive groups; namely, we can analyze disparities in: admissions rates ($\frac{\sum_t E_{g,t}^U}{\sum_t A_{U,t}^g}$); graduations rates for undergraduate, Master's and doctoral programs ($\frac{\sum_t G_{g,t}^U}{\sum_t C_{g,t}^U}$, $\frac{\sum_t G_{g,t}^M}{\sum_t C_{g,t}^M}$, and $\frac{\sum_t G_{g,t}^D}{\sum_t C_{g,t}^D}$); and salaries ($\frac{\sum_t S_{g,t}}{\sum_t W_{g,t}}$). Hence, the indicators

Table 7: `MAFE-Edu` Component Indicators

| Indicator | Description |
|---|---|
| $P_t$ | Employer Profits at time step $t$ |
| $A_{U,t}^g$ | Number of people that applied to university from Group $g$ at time step $t$ |
| $E_{g,t}^U$ | Number of students that entered university from Group $g$ at time step $t$ |
| $C_{g,t}^U$ | Initial number of students in undergraduate class currently graduating from Group $g$ at time step $t$ |
| $G_{g,t}^U$ | Number of students that graduated from undergraduate program from Group $g$ at time step $t$ |
| $C_{g,t}^M$ | Initial number of students in undergraduate class currently graduating from Group $g$ at time step $t$ |
| $G_{g,t}^M$ | Number of students that graduated from master's program from Group $g$ at time step $t$ |
| $C_{g,t}^D$ | Initial number of students in undergraduate class currently graduating from Group $g$ at time step $t$ |
| $G_{g,t}^D$ | Number of students that graduated from doctoral program from Group $g$ at time step $t$ |
| $W_{g,t}$ | Number of people in the workforce from Group $g$ at time step $t$ |
| $S_{g,t}$ | Sum of all salaries of people in workforce from Group $g$ at time step $t$ |

provided by the environment at each time step are used to measure five rewards and five fairness disparities.

**Mathematical Modeling:**

Student GPA Dynamics:

We model a student's cumulative at time step $t$, $\text{GPA}_t$, as a random process given by the following recursion:

$$\text{GPA}_t = \frac{(t-1)\text{GPA}_{t-1} + \widehat{\text{GPA}}_t}{t}, \tag{18}$$

where $\widehat{GPA}_t$ represents a student's semester GPA at time step $t$. We model $\widehat{\text{GPA}}_t$ as being a noisy estimate of the student's previous semester GPA, $\widehat{\text{GPA}}_{t-1}$, assuming that the GPA that the student most recently received is most indicative of the trajectory of their performance in classes. That is,

$$\widehat{\text{GPA}}_t = \widehat{\text{GPA}}_{t-1} + \epsilon, \tag{19}$$

where $\epsilon \sim \text{Uniform}[-\Delta, \Delta]$ for some constant $\Delta$.

The final critical ingredient required for completing the modeling of a student's GPA is to determine how to set $\widehat{\text{GPA}}_0$, the initial condition for Equation 18. For this task, we model $\widehat{\text{GPA}}_0$ as a noisy function of the subset of an individual's feature vector, $\mathbf{u} \subset \mathbf{v}$, containing features from the IPUMS EDUC dataset given by:

$$\widehat{\text{GPA}}_0 = f(\mathbf{u}) + \gamma_0 + \gamma_1 \cdot (1 - \text{ANNUALTUIT})$$
$$+ \gamma_2 \cdot \text{INMINTYPGRM} \tag{20}$$

We obtain $f$ through training a regressor using the samples available in the IPUMS EDUC dataset where all IPUMS EDUC features from Table 6 are treated as the independent variables and GPA is treated as the dependent variable. We particularly use ridge regression for this task. $\gamma_1$ and $\gamma_2$ are user-specified weights that introduce the effect that student supports provided by the University Budget Allocation Agent have on improving student progress through the university. For these terms, we assume that ANNUALTUIT is normalized to be a percentage (between 0 and 1) and INMINTYPGRM is a binary valued variable. $\gamma_0 \sim$ Uniform$[-\delta + C, \delta + C]$ is used to introduce stochasticity in baseline GPAs and is represented by uniform random noise over a window of length $2\delta$. $C$ centers this window and is

adjusted based on the academic supports provided to as student. If an individual receives a significant scholarship or is provided an academic mentor, then $C > 0$. Otherwise, $C = 0$. Taken collectively, $f$ represents measures an individual's baseline academic merits, while $V$ represents intervention adjusted uncertainty in an individual's performance.

Likelihood of Leaving College:

When deciding whether remaining enrolled in school is beneficial, a student must way a variety of factors, his/her performance thus far, the tradeoff in time that could be spent elsewhere, and the price paid for tuition. Thus, we obtain the likelihood that an individual leaves college at time step $t$ through sampling Bernoulli distribution, Bernoulli($P_t^{Leave}$), where $P_t^{Leave}$ is given by:

$$P_t^{Leave} = \sigma(\alpha_0 + \alpha_1 \text{GPA}_t + \alpha_2 \text{ANNUALTUIT}_t$$
$$+ \alpha_3 \text{TIMEINUNIV}_t + \alpha_4 \text{TIMEINUNIV}_t^2). \tag{21}$$

$\text{GPA}_t$ and $\text{ANNUALTUIT}_t$ are modeled as linear functions with negative and positive effects, respectively, on a student's likelihood of leaving college. Therefore, we assume $\alpha_1 < 0$ and $\alpha_2 > 0$.

We represent the effect of enrollment duration on the likelihood of departure using an inverted quadratic function, reflecting the intuition that students are less likely to leave immediately after enrolling. Consequently, $\alpha_3 < 0$ and $\alpha_4 > 0$.

The rationale is as follows: During the initial period after enrollment, students may be more inclined to leave if their academic performance is poor or their expectations are unmet. However, as time progresses, the likelihood of departure decreases. This is because students invest increasing resources into their degree and draw closer to completion, making withdrawal less advantageous.

Finally, note that tuition is influenced by the amount of scholarship funding provided by the university.

Student-Teacher-Infrastructure Ratio:

As previously discussed, we assume that the university's ability to provide quality instruction to students is limited by the number of students it can enroll at any given time. This enrollment cap is dependent on the size of the faculty. However, the number of faculty members that can be supported on campus is in turn limited by the availability of infrastructure, such as classrooms, offices, and laboratories, which are necessary for both faculty research and instruction. Therefore, the number of faculty members and the available student seats on campus are both determined by the amount of infrastructure the university has.

Specifically, the number of faculty members supported by the university at time $t$ is linearly proportional to the amount of infrastructure available. Similarly, the student enrollment capacity at any time is also linearly proportional to the infrastructure available. To align with common intuition, we set the proportionality constants governing faculty size and student enrollment to values significantly greater than one. This reflects the fact that multiple faculty members can occupy a single building, and many students are taught by a single faculty member. The ratio between the student enrollment capacity and the number of faculty indicates the student-to-faculty ratio, with larger ratios corresponding to larger class sizes.

Cost of Building University Infrastructure:

By university infrastructure, we refer to all construction (including classrooms, laboratories, offices, etc.) that must take place to increase the student and faculty population capacities on a university campus. We use the same equations used to the model cost of building new hospital infrastructure here for building new university infrastructure, though the interpretation is changed. That is, building new infrastructure involves two main costs: a base cost, which is incurred for any construction plan, and a proportional cost, which depends on the number of new university infrastructure units built. The total cost of building new infrastructure is modeled as a linear function, where the base cost is added to the cost that increases with the number of new beds. This creates a trade-off for the university budget allocator planner, who must decide when to invest in infrastructure. Investing in small

projects repeatedly can become expensive due to the base cost, while waiting to fund a larger project may limit the number of students the university can admit.

Notably, counter to the hospital MAFE, in the university MAFE, we also assume that building new university infrastructure comes with an additional recurring cost which represents then additional salaries for faculty and staff that are supported by the addition of this new infrastructure.

Time to Build University Infrastructure:

We model the time to build university infrastructure identically to cost of hospital infrastructure, but with a different interpretation. Specifically, the time required for construction depends on the size of the project. There is a base amount of time required for planning and setting up the project, and additional time required is linearly proportional to the number of new beds added by the project.

An Individual's Utility to An Employer:

An employee's value to an employer may depend on a variety of factors that comprise his/her merits, including his/her years of experience, level of degree attainment, cumulative GPA, the salary he/she receives, and whether or not his/her hiring affects an employer's diversity incentives. Moreover, these factor may interact, making modeling the effect that they have on the profits made by an employer non-linear and thus more complicated. With this in mind, we model the profits an employee brings to an employer at time step $t$ using an inverted quadratic function of a person's salary, $SALARY_t$:

$$U(\text{SALARY}_t) = \alpha_0 + \alpha_1(\text{SALARY}_t + \text{DIVINVEST}_t)$$
$$- \alpha_2 \text{SALARY}_t^2, \tag{22}$$

where $\alpha_0$ and $\alpha_1 > 0$ are user-defined parameters and $\alpha_2$ is a function of a person's cumulative college, $GPA$; the level of a persons highest degree attained, $DEGREE$; and the number of years of experience a person has working, $EXPERIENCE_t$. That is, $\alpha_2$ takes the following form with user defined parameter's $\beta_0, ..., \beta_3$:

$$\alpha_2 = \beta_0 + \beta_1 \text{GPA}_t + \beta_2 \text{TIMEINUNIV}_t$$
$$+ \beta_3(\text{EXPERIENCE}_t - \text{EXPERIENCE}_t^2) \tag{23}$$

To ensure that Equation 22 takes an inverted quadratic form, The parametrization of Equation 23 must be selected so that $\alpha_2 > 0$.

The intuition behind the design of Equation 22 is as follows. An increase in employee income leads to a marginal improvement in productivity, which directly benefits employer profits. This positive relationship is captured by the linear term in Equation 22. On the other hand, paying an employee a higher salary also directly reduces the employer's profits, which is modeled by the negative quadratic term in the same equation. The balance between these two effects depends on the interactions between employee salary and other factors captured by $\alpha_2$. The coefficients $\beta_0, \ldots, \beta_3$ can be adjusted to reflect the relative influence of these factors on employer profits. We set these values based on the intuition that higher education and better educational performance justify higher wages for employees, as they are likely to increase productivity. The quadratic term for experience captures the dual effects of greater experience: while more experience may enhance job performance, it could also lead to less flexibility in work habits and reduced exposure to the latest industry developments, as newer educational techniques and trends are typically acquired earlier in a career.

Effect of Tertiary Investment:

We use the same modeling as was performed to model the effect of public investment in Section G to model the effect of tertiary investment for `MAFE-Edu`, just with different application interpretation. Namely, in each time step, a subset of the updateable features in $\mathbf{v}_v$ associated with each individual in Region $g$ will improve with probability $P_{g,t}^{improve}$, remain unchanged with constant probability $U$, or deteriorate with probability $1 - P_{g,t}^{improve} - U$. We treat $U$ as a user specified constant. The value of $P_{g,t}^{improve}$ is affected by the amount of the Central Planner's budget that is used on tertiary investment in in Region $g$ at time step $t$.

Table 8: Experimental Hyperparameters

| MAFE | | |
|---|---|---|
| MAFE-Loan | MAFE-Health | MAFE-Edu |
| *Episode Initialization Parameter* | | |
| Time Horizon (T): 400
Action frequency ($k$) for agents ($\pi_1, \pi_2, \pi_3$): (1,1,1)
Sensitive attribute include as feature: Yes
Equation Parameters | Time Horizon (T): 100
Action frequency ($k$) for agents ($\pi_1, \pi_2, \pi_3$): (6,1,6)
Sensitive attribute include as feature: Yes
Equation Parameters
Planner Budget ($\hat{B}$): 2.5e8
Number of Geographic Regions ($N_g$): 4 | Time Horizon (T): 100
Action frequency ($k$) for agents ($\pi_1, \pi_2, \pi_3, \pi_4$): (1,1,1,1)
Sensitive attribute include as feature: Yes
Equation Parameters
Planner Budget ($\hat{B}$): 2.5e7
Number of Geographic Regions ($N_g$): 9 |
| *F-MACEM Training Parameters* | | |
| Elite set size ($p\%$): 0.2
Epochs: 40
Episodes Per Epoch: 100 | Elite set size ($p\%$): 0.2
Training Epochs: 40
Training Epochs: 100 | Elite set size ($p\%$): 0.2
Training Epochs: 40
Training Epochs: 100 |
| *Reward/Fairness Measure Normalization Factors for Frontier Results* | | |
| Bank Profits: 8.9e4
Admissions Rate: N/A
Admissions Rate Disparity: N/A
Wait Time Disparity: Sum of Average Wait Times
Default Rate: N/A
Default Rate Disparity: N/A | Insurance Profits: 7.2e8
Insurance Rate: N/A
Mortality Rate: N/A
Incidence Rate: N/A
Insurance Rate Disparity: N/A
Mortality Rate Disparity: N/A
Incidence Rate Disparity: N/A | Employer Profits: 6.0e5
Default Rate: N/A
Admissions Rate: N/A
Graduation Rate: N/A
Salary Disparity: Sum of Average Salaries
Admissions Rate Disparity:N/A
Graduation Rate Disparity:N/A |
| *Mathematical Modeling Parameters* | | |
| Equation (10): $\mu = 0, \sigma = 0.025$

Equation (11): 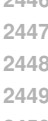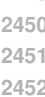 $\mathbf{c} = \begin{bmatrix} c_{FICO\_LOW} \\ c_{FICO\_HIGH} \\ c_{mths\_since\_last\_delinq} \end{bmatrix} = \begin{bmatrix} 100 \\ 100 \\ 5 \end{bmatrix}$ | Equation (12): $A = B = 0.4$
Equation (15):
$\quad C = 0, D = 1.03, E = -7, F = 0, G = 0, H = 0$
$\quad$ (For $P^{Terminate}$)
$\quad C = 1.96, D = -1.02, E = 3, F = 3, G = 3, H = -7$
$\quad$ (For $P^{Mortality}$)

Cost of Hospital Infrastructure:
$\quad$ Base Cost=3e7
$\quad$ Proportional Cost=1e6

Time to Build Hospital Infrastructure:
$\quad$ Base Time=0.5
$\quad$ Proportional Time=2

Equation (18): $Q = 0.29, R = 0.4, V = \frac{16 \cdot N_g}{\hat{B}}, W = 4$

Equation (19): $U = 0.2$ | Equation (21): $\Delta = 0.25$

Equation (22): $\gamma_1 = 0.1, \gamma_2 = 0.3, \delta = 0.4$

Equation (23):
$\alpha_0 = 0, \alpha_1 = -1, \alpha_2 = 0.5, \alpha_3 = -0.05, \alpha_4 = 0.001$
$\quad$ (For Undergraduate Degree)
$\alpha_0 = 0, \alpha_1 = -1, \alpha_2 = 0.5, \alpha_3 = -0.05, \alpha_4 = 0.001$
$\quad$ (For Master's Degree)
$\alpha_0 = 0, \alpha_1 = -1, \alpha_2 = 0.5, \alpha_3 = -0.05, \alpha_4 = 0.001$
$\quad$ (For Doctoral Degree)

Student-Teacher-Infrastructure Ratio: $1 : 5 : 75$

Cost of Building University Infrastructure:
$\quad$ Base Cost=1e6
$\quad$ Proportional Cost=1e6
Time to Build University Infrastructure:
Equation (24): $\alpha_0 = 0.1, \alpha_1 = 1.2$
Equation (25): $\beta_0 = 3, \beta_1 = -1.1, \beta_2 = -1.1, \beta_3 = -1.1$
Equation (26): $Q = 0.39, R = 0.4, V = \frac{16 \cdot N_g}{\hat{B}}, W = 4$
Equation (27): $U = 0.2$ |

In particular, we model $P_{g,t}^{improve}$ as a function of the amount of the planners budget invested in the region in which this individual is located at time $t$. For constant hyperparameters $Q, R, V,$ and $W$, this is given by the following equation:

$$P_{g,t}^{improve}(x) = Q + R\sigma(V \cdot x + W) \tag{24}$$

where $\sigma$ represents a sigmoid function. We assume this equation is tuned so that $P_{g,t}^{improve}$ is non-negative and

$$\sup_x P_{g,t}^{improve}(x) + U = 1. \tag{25}$$

To determine if an individuals features improve, deteriorate, or remain unchanged we sample a uniform distribution over the range $[0, 1]$ and update the features appropriately based on the segment in which the output value lands—$[0, P_{g,t}^{improve}]$, $(P_{g,t}^{improve}, P_{g,t}^{improve} + U]$, or $(P_{g,t}^{improve} + U, 1]$.

**Episode Termination:** An episode may terminate for three reasons. First, if the agents produce actions that lead them to successfully reach the user specified terminal time step, the episode terminates. Conversely, the environment may also terminate early if any entity in the institution fails. Particularly, if the university is ever unable to support the salaries of its staff and faculty due to improper allocation of its budget or a lack of enough money in the budget. An episode may also fail if net profits accumulated by the employer agent are ever negative.

## I  HYPERPARAMETERS

In this section, we provide a full list of the parameters we selected for conducting the experiments presented in this paper. These values are organized in Table 8.

## J  Time and Space Complexity

In all experiments, we train F-MACEM for 40 epochs. During each epoch, 100 episodes are run using different parameter samples of a multi-layer perceptron (MLP). The networks used are shallow, consisting of only two layers each. Only the parameters and objective function values are stored during training to perform elite optimization updates based on the elite set from each epoch (see Appendix C for algorithm details). As a result, the algorithm has low memory requirements. The time complexity for running 40 epochs of 100 episodes, each with 400 time steps, varies depending on the environment due to differences in state transition dynamics. Training typically takes 2-3 days per run. However, multiple runs can be executed in parallel on a system with an Intel(R) Core(TM) i9-9900 CPU @ 3.10GHz, thanks to the algorithm's low storage demands.

For completeness, we provide the table below to document the average per–time-step cost (in seconds) of taking a step inside each MAFE environment over 12,000 steps.

Table 9: Average per–time-step cost across MAFE environments.

| Environment | Avg. Time per Step (s) | Std. Dev. (s) |
|---|---|---|
| MAFE-Loan | 0.13 | 0.05 |
| MAFE-Edu | 0.40 | 0.02 |
| MAFE-Health | 0.25 | 0.02 |

Model update costs are consistent across environments ($< 0.04$ seconds per update) and occur only once per epoch (after 100 episodes), making them negligible relative to per-step environment costs.

