# OpenReview forum: "MAFE: Enabling Equitable Algorithm Design in Multi-Agent Multi-Stage Decision-Making Systems"
_ICLR.cc/2026/Conference — Submitted to ICLR 2026_

### Official Review · Reviewer_YFah · 2025-10-29

**Soundness:** 3
**Presentation:** 3
**Contribution:** 3
**Rating:** 6
**Confidence:** 3

**Summary:**

This paper proposes the Multi-Agent Fair Environments (MAFE) that provides a fairness benchmark for multi-agent sequential decision-making systems. They mention that previous work mostly focus on static or single-agent settings, but most real-world scenarios are multi-agent. They offer 3 domains: loan processing, healthcare, and higher education, and offer heterogeneous agents. They propose the Fair Multi-Agent Cross Entropy Method (F-MACEM) method to jointly optimize over agents for reward and fairness components. They use real-world data to compute priors over the domains in consideration.

**Strengths:**

* The study of long-term fairness in multi-agent sequential decision making is an important topic.
* The proposed domains use real-world data as part of the simulation.
* The empirical results look good using their proposed method, and the benchmark methodology seems to make sense.

**Weaknesses:**

* The paper advertises a benchmark that promotes reproducibility and compatibility with standard MARL libraries. However, I do not see any experiments showcasing this. There are F-MAPPO and F-MADDPG, but the authors do not specify whether these come from off-the-shelf libraries. I think the paper would benefit from including comparisons of more baseline off-the-shelf MARL algorithms.
* While the use of real-world datasets to instantiate the MAFEs is nice, there is limited discussion on how realistic the environment dynamics portray real-world observations. I do not recall seeing this listed as a limitation anywhere.
* I find it strange that the F-MACEM method is proposed in this paper but its analysis appears limited and all discussion of it is moved to Appendix. What is its novelty, how does it compare with more existing algorithms, including those more similar to it that e.g. uses evolutionary methods?
* The related works, especially Sec. 2.2, can benefit from more analysis/citations and what sets your work apart. For example, [1, 2].
* The linear combination of reward and fairness terms as an augmented reward yields a potentially nontrivial multi-objective optimization problem which may be hard to select appropriate weights and directly solve.


Minor: Equation 1 of Section 5, I think $\theta_n, \alpha, \beta$ not defined.

[1] Aloor, J. J., Nayak, S. N., Dolan, S., & Balakrishnan, H. (2024). Cooperation and Fairness in Multi-Agent Reinforcement Learning. ACM Journal on Autonomous Transportation Systems, 2(2), 1–25. doi:10.1145/3702012
[2] Ju, P., Ghosh, A., & Shroff, N. B. (2023). Achieving Fairness in Multi-Agent Markov Decision Processes Using Reinforcement Learning. arXiv [Cs.LG]. Retrieved from http://arxiv.org/abs/2306.00324

**Questions:**

Please see weaknesses. Some additional questions:
* In Figure 3, what baseline is used? How are interventions done?
* Is there a way to specify soft/hard constraints in this framework? Suppose some fairness objectives are extremely important, so the agent must satisfy them. Simply setting a very high weight on it may yield an overly conservative policy. Let me know if I missed this discussion anywhere in the paper.

---

> ### Author Response · Authors · 2025-11-23
> **Response to Reviewer (Part 1)**
>
> We thank Reviewer $YFah$ for the thoughtful evaluation and for **highlighting the importance of multi-agent long-term fairness, the value of grounding the domains in real-world data, and the strength of our empirical results and benchmarking methodology.**
>
> Overall, the reviewer’s concerns center on clarifying the benchmark’s compatibility with standard MARL libraries, the realism of environment dynamics, and the scope and purpose of the F-MACEM adaptation. We have expanded the discussion of library compatibility, added clarification regarding realism and limitations, and more clearly positioned F-MACEM as a representative evolutionary baseline rather than a primary algorithmic contribution. We also refined related-work positioning and clarified the treatment of multi-objective fairness–reward combinations.
>
> We address **all** concerns below.
>
> ---
>
> > The paper advertises a benchmark that promotes reproducibility and compatibility with standard MARL libraries. However, I do not see any experiments showcasing this. There are F-MAPPO and F-MADDPG, but the authors do not specify whether these come from off-the-shelf libraries. I think the paper would benefit from including comparisons of more baseline off-the-shelf MARL algorithms.
>
> **MAFE supports standard MARL algorithms with fairness extensions.**
> * **Clarification:** MACEM, MAPPO, and MADDPG are **standard, existing MARL algorithms**. We adapt them to be **Fairness-Aware** (F-MACEM, F-MAPPO, F-MADDPG).
> * **Adaptation Rationale:** Given the complexity of our environments (realistic, population-level, multi-stage), adapting these standard algorithms is non-trivial and represents an important research direction, showcasing MAFE's compatibility.
> * **Representativeness:** F-MAPPO and F-MADDPG provide valuable policy-gradient-based perspectives complementary to the evolutionary F-MACEM.
>
> > While the use of real-world datasets to instantiate the MAFEs is nice, there is limited discussion on how realistic the environment dynamics portray real-world observations. I do not recall seeing this listed as a limitation anywhere.
>
> **We provide qualitative evidence of realism, with plans for quantitative validation.**
> * **Current State:** While we lack quantitative realism evaluation, Appendix D.5 provides **qualitative evidence** that learned agents balance incentives (e.g., premiums, subsidies) plausibly, reflecting real-world trade-offs (fairness, financial stability).
> * **Limitation & Future Work:** Quantitative assessment requires real-world implemented policy data. We plan to extend MAFE-Health to incorporate and validate against accessible real-world dynamics data (e.g., OxCGRT COVID-19 Government Response Tracker Dataset).
>
> > I find it strange that the F-MACEM method is proposed in this paper but its analysis appears limited and all discussion of it is moved to Appendix. What is its novelty, how does it compare with more existing algorithms, including those more similar to it that e.g. uses evolutionary methods?
>
> **F-MACEM serves as a representative evolutionary baseline, not a primary novel contribution.**
> * **Purpose:** F-MACEM is an **adaptation** of the Cross-Entropy Method (CEM) to evaluate the **learnability** of fairness-aware objectives within the proposed MAFEs.
> * **Limited Discussion Rationale:** Due to ICLR page constraints, the detailed description and analysis were placed in the Appendix.
> * **Update:** We have clarify F-MACEM's role as an adaptation of a representative evolutionary algorithm and cite well established comparative research on CEM [1,2,3] for completeness.
>
> [1] Szita, I., & Lörincz, A. (2008). Online variants of the cross-entropy method. arXiv preprint arXiv:0801.1988. https://arxiv.org/abs/0801.1988
>
> [2] Banks, C., Gennet, N., & Coogan, S. (2022). LTL Cross Entropy Optimization for Quadcopter Task Orchestration. Proceedings of the IEEE International Conference on Robotics and Automation. https://coogan.ece.gatech.edu/papers/pdf/banks2021ltl.pdf
>
> [3] Piroddi, L., & Borghini, F. (2021). A proposal for a cooperative cross-entropy method to tackle the unit commitment problem. Computers & Industrial Engineering, 162, 107764. https://dl.acm.org/doi/10.1016/j.cie.2021.107764

---

> > ### Author Response · Authors · 2025-11-23
> > **Response to Reviewer (Part 2)**
> >
> > > The related works, especially Sec. 2.2, can benefit from more analysis/citations and what sets your work apart. For example, [1, 2].
> >
> > **MAFE provides a realistic benchmark that complements existing algorithmic work on fairness.**
> >
> > * **Expansion:** We have expand Section 2.3 to include additional analysis and citations (e.g., Aloor et al., Ju et al.).
> > * **Differentiation:** Prior work focuses on simplified environments or theoretical treatment. Our contribution is the creation of **realistic, multi-agent environments** instantiated with real-world data heterogeneity and complex interactions, serving as a **benchmark** for evaluating fairness-aware MARL algorithms. Specifically, [1] considers a simplified navigation environment with randomly distributed, homogeneous agents that are not instantiated from real-world data. In contrast, our environments incorporate realistic, data-driven heterogeneity and complex inter-agent interactions. [2], on the other hand, provides a primarily theoretical treatment of fairness in multi-agent Markov decision processes, focusing on algorithmic guarantees rather than practical, empirical evaluation.
> >
> >
> > > The linear combination of reward and fairness terms as an augmented reward yields a potentially nontrivial multi-objective optimization problem, which may be hard to select appropriate weights and directly solve. Is there a way to specify soft/hard constraints in this framework? Suppose some fairness objectives are extremely important, so the agent must satisfy them. Simply setting a very high weight on it may yield an overly conservative policy. Let me know if I missed this discussion anywhere in the paper.
> >
> > **The framework supports exploring the trade-off, and we view constraints as future work.**
> > * **Current Framework:** We analyze the fairness-accuracy trade-off, presenting the **Pareto frontiers** (Appendix D.4, Figure 9) to illustrate how weight choices impact performance.
> > * **Future Work:** We agree that extending the framework to include explicit constraint-based formulations (soft/hard) is an important direction for future research.
> >
> >
> > > In Figure 3, what baseline is used? How are interventions done?
> >
> > **The baseline is the outcome *without* the intervention applied.**
> > * **Clarification:** The baseline curves in Figure 3 represent the system's outcomes when the specific intervention used for the red curve (the intervention of interest) is **not applied at all**. We have clarified this in the manuscript.
> >
> >
> > > Minor: Equation 1 of Section 5, I think not defined.
> >
> > **Variables in Equation 1 are now defined in the manuscript.**
> > * **Definition:** We clarified that $\theta_n$ are the parameters of the model used to produce the action taken by agent $n$; $\beta_m$ is a user-defined weight for the $m$-th fairness penalty; and $\alpha_k$ is a user-defined weight for the $k$-th reward.

---

> ### Author Response · Authors · 2025-11-26
>
> Dear reviewer $YFah$,
>
> We have updated the paper with clarifications, carefully responded to your comments, and believe that all outstanding questions have been resolved. We hope this will address your concerns, and you may consider the paper suitable for a potentially higher score. We remain committed to further clarifying any remaining points, should any remain unclear. We sincerely appreciate the time and effort invested by you and the Area Chairs throughout this process.

---

> > ### Comment · Reviewer_YFah · 2025-11-26
> >
> > Thank you for the detailed rebuttal, my concerns have been addressed. I still find it strange to have the benchmark as the primary contribution, then propose a new algorithm to conduct analysis with only brief introduction in the appendix.
> >
> > Since my original score is already in the accept range, I will keep my score.

---

### Official Review · Reviewer_qhsr · 2025-10-31

**Soundness:** 1
**Presentation:** 1
**Contribution:** 2
**Rating:** 2
**Confidence:** 5

**Summary:**

This paper introduces MAFE (Multi-Agent Fair Environments), a benchmarking suite designed to evaluate fairness-aware policies in multi-agent reinforcement learning settings. It provides three domain-specific environments: MAFE-Loan (financial lending), MAFE-Health (healthcare), and MAFE-Edu (education), and demonstrates their utility through experiments using an evolutionary strategy adapted for fairness-aware optimization.

**Strengths:**

1. I acknowledge that there is currently a gap in the community for more sophisticated benchmarks for measuring fairness in dynamic environments, and this paper addresses that need.

2. The modeling details, particularly those based on real data, represent substantial extensions over existing benchmarks such as D’Amour et al.

**Weaknesses:**

1. The proposed algorithm is a straightforward adaptation of the cross-entropy method, and the environments are not benchmarked against existing fairness-aware RL algorithms cited by the authors. While algorithmic novelty is not essential for a benchmark paper, I believe the environments should still be evaluated using state-of-the-art fairness-aware RL algorithms to better demonstrate their utility. However, such comparison is completely missing from the paper and the environments are benchmarks on two standard MARL algorithms.

2. From my understanding, all environments assume cooperation among agents. This is a major limitation, as such full cooperation is rarely the case in reality. Since the paper’s goal is to provide realistic benchmark environments, I see this as a significant issue. In addition, it would be valuable for these environments to include a single-agent variant, where other agents are controlled by the environment through logic or heuristics. This would increase the impact of the work by allowing the benchmark to be used for both single-agent and multi-agent research.

3. The reported runs using an MLP, 100 episodes with 400 steps each (a total of 40k steps), take 2–3 days to complete. This seems surprisingly long, given that the simulations are based on relatively simple mathematical equations (with no rendering or complex physics, unlike many existing benchmarks). This could significantly limit the benchmark’s practicality and adoption. Most RL algorithms typically require 500k–1M steps for training, which would take more than a month at this rate!! The authors should optimize the runtime or provide simplified versions that can be completed in less than a day using an MLP model, if their goal is to have their benchmark be used by the community.

4. The benchmark imposes certain unnecessary constraints on the algorithm and policy. For example, it requires the policy to be permutation equivariant. Such requirements add unnecessary complexity and reduce the benchmark’s practical usability. A good benchmark should be model agnostic and should not impose specific requirements like this on the algorithm.

**Minor:** Many of the citations show the authors twice, like Atwood et al. Atwood et al. (2019).


**Overall Assessment:** Aside from the weaknesses mentioned above, my main concern is whether this work is suitable for the main conference track, given its limited algorithmic contribution. I believe it would be a much better fit for a dataset/benchmark track, although such a track does not currently exist at this conference.

**Questions:**

1. Why are the policies required to be permutation equivariant? How is this requirement imposed in the benchmark? If an algorithm that is not permutation equivariant outperforms one that is, what would that imply for the benchmark?

2. Why do the runs take 2–3 days for only 40k environment steps? Could you provide profiling of how much time is spent on environment simulation versus the algorithm itself?

3. What happens if agents have conflicting objectives and cannot cooperate? Can the benchmark handle such non-cooperative scenarios?

---

> ### Author Response · Authors · 2025-11-23
> **Response to Review (Part 1)**
>
> We thank Reviewer $qhsr$ **for acknowledging the gap in sophisticated dynamic fairness benchmarks and for recognizing that our modeling—particularly its grounding in real data—represents a substantial extension over existing work.**
>
> Overall, **the reviewer’s concerns largely stem from misunderstandings about the generality of the framework, the intended scope of the benchmark contribution, and the efficiency and flexibility of the environments.** We clarify that MAFE fully supports non-cooperative settings, is compatible with additional fairness-aware RL algorithms, and does not impose restrictive architectural requirements. We also provide clarification on runtime, single-agent variants, and model-agnostic design choices.
>
> We address **all** concerns below.
>
> ---
>
> > Aside from the weaknesses mentioned above, my main concern is whether this work is suitable for the main conference track, given its limited algorithmic contribution. I believe it would be a much better fit for a dataset/benchmark track, although such a track does not currently exist at this conference.
>
> **Suitability for Main Conference Track (Algorithmic Contribution vs. Benchmark)**
> * **The work is suitable for the main conference track under the established 'Datasets and Benchmarks' theme.**
> * **Justification:** The Call for Papers for ICLR 2026 https://iclr.cc/Conferences/2026/CallForPapers includes **Datasets and Benchmarks** as a specific submission area, aligning perfectly with our contribution.
> * **Contribution:** Our paper introduces a novel suite of \textbf{Multi-Agent Fair Environments (MAFE)} designed to simulate realistic, modular, and dynamic systems. This work fills a critical gap in the fairness community by providing realistic benchmarks for comprehensive empirical fairness evaluation in societal domains.
> * **Scope:** Specifically, our research:
>   * Comprehensively models three distinct societal domains at an intermediate level of abstraction.
>   * Adapts multiple popular multi-agent algorithms (F-MACEM, F-MAPPO, F-MADDPG) to operate within these environments.
>   * Demonstrates the effectiveness of the environment for novel research and encourages widespread adoption.
>
> > I believe the environments should still be evaluated using state-of-the-art fairness-aware RL algorithms to better demonstrate their utility.
>
> **Evaluation with State-of-the-Art Fairness-Aware RL Algorithms**
> * We agree that evaluation with additional state-of-the-art fairness-aware RL algorithms would further demonstrate MAFE's utility.
> * **Current Approach Justification:** We selected F-MACEM, F-MADDPG, and F-MAPPO because they:
>   * Are built on **standard, established RL algorithms** (CEM, DDPG, PPO) with known implementations.
>   * Capture **diverse perspectives** in modern MARL (evolutionary vs. policy-gradient).
>   * Required **non-trivial adaptation** to handle the multi-agent fairness objectives and the complexity of MAFEs.
> * **Future Work:** We view extending our evaluation to more recent fairness-aware RL algorithms as an **important direction for future work**. As demonstrated in Appendix D.6, algorithm--environment compatibility critically influences performance in MAFEs, meaning direct adaptation of other specialized fairness-aware RL algorithms is a non-trivial research task in itself.
>
> > From my understanding, all environments assume cooperation among agents. This is a major limitation, as such full cooperation is rarely the case in reality. What happens if agents have conflicting objectives and cannot cooperate? Can the benchmark handle such non-cooperative scenarios?
>
> * **Current experiments use cooperative objectives for clarity, but the framework is general-purpose**. We re-emphasized the following points:
>   * The current experiments use cooperative objectives for simplicity to allow for focusing on the main setup and operation of the MAFE framework.
>   * The framework supports arbitrary agent reward functions, heterogeneous objectives, and asymmetric information structures.
>   * Future research can build on MAFE to study richer strategic settings.

---

> ### Author Response · Authors · 2025-11-23
> **Response to Review (Part 2)**
>
> > Why do the runs take 2–3 days for only 40k environment steps? Could you provide a profiling of how much time is spent on environment simulation versus the algorithm itself?
>
> * **The runs do not use 40k steps; this is a misunderstanding.** The reviewer appears to have interpreted 100 episodes over 40 epochs as yielding 40,000 environment steps.  However, this calculation ignores that each episode comprises hundreds of time steps. For example, in our experiments, MAFE-Loan uses 400-step episodes, which results in 400 x 100 x 40 = 1,600,000 environment steps, not 40,000.
> * **Reporting cost profiling.** For MAFE-Loan, the average per–time-step runtime is 0.05 seconds (standard deviation 0.03) over 12,000 steps. We are currently generating the per–time-step profiles for the remaining environments and will share the complete results soon.
> * **Episode length and training horizon are user-tunable hyperparameters**. The choice of horizon depends on the temporal dynamics of each domain. Since this is a hyperparameter, users can shorten episodes or reduce the number of episodes per epoch to reduce the runtime of an episode, though this may come at the cost of capturing fewer long-horizon effects.
> * **There is substantial room for efficiency improvements using modern acceleration frameworks.** Our implementation prioritizes clarity and reproducibility on a standard CPU (Intel Core i9-9900 @ 3.10GHz). Frameworks such as JAX, which support just-in-time compilation and vectorized computations, can significantly reduce per-step simulation cost in future implementations without modifying the environments themselves.
>
> > The benchmark imposes certain unnecessary constraints on the algorithm and policy. For example, it requires the policy to be permutation equivariant. Such requirements add unnecessary complexity and reduce the benchmark’s practical usability. A good benchmark should be model agnostic and should not impose specific requirements like this on the algorithm. Why are the policies required to be permutation equivariant? How is this requirement imposed in the benchmark? If an algorithm that is not permutation equivariant outperforms one that is, what would that imply for the benchmark?
>
> * **The framework itself does not require permutation-equivariant architectures; any algorithm can be used**. MAFE supports both set-structured observations (as in our implementation) and fully aggregated or summary-based representations. The mathematical framework places no restrictions on the function class of the policy; our use of permutation-equivariant networks is an implementation choice aligned with the social semantics of the domains.
> * **Permutation equivariance is not a benchmark requirement; it reflects the semantics of the observation space**. In MAFE, each row of the observation corresponds to a specific individual (patient, borrower, student), and the ordering of individuals has no meaning. Policies should behave identically if rows are permuted, since no person’s features have changed.
> * **Permutation equivariant architectures are therefore an appropriate design choice for variable-sized populations**. We implement equivariant variants of MADDPG, MAPPO, and F-MACEM because they naturally operate on unordered sets of individuals whose size and composition change over time. This mirrors standard practice in set- and graph-based MARL/RL.
> * **If a non-equivariant model outperforms an equivariant one, this would indicate reliance on ordering artifacts rather than meaningful signal.** Such performance would reflect exploitation of representational quirks (e.g., reacting differently when two individuals swap positions) rather than better decision-making. Equivariant models avoid this issue and yield fair, stable, and interpretable comparisons.

---

> ### Comment · Reviewer_qhsr · 2025-11-23
>
> Thank you for your rebuttal.
>
> While some concerns were partially addressed, the main issues remain:
>
> * The single-agent variant is deferred to future work, but I believe it is essential for the first version.
> * Comparisons with fairness-aware baselines are also pushed to future work, which I see as a significant gap.
> * The relevance to ICLR is still unclear given the limited benchmarking and algorithmic development.
> * Could you clarify what “epoch” means in your setting? As I also wrote in my original review, I calculated 40k steps as 100 episodes × 400 steps per episode. Typically, an epoch refers to repeatedly training on a fixed dataset; in online RL, the notion of epochs is uncommon and step counts are generally used instead.
>
> Given these points, I am keeping my score as is.

---

> > ### Author Response · Authors · 2025-11-25
> > **MAFEs Cost Profiling**
> >
> > For completeness, we provide the table below to document the average per–time-step cost (in seconds) of taking a step inside each MAFE environment over 12,000 steps.
> >
> > | Environment   | Avg. Time per Step (s) | Std. Dev. (s) |
> > |---------------|-------------------------|----------------|
> > | MAFE-Loan     | 0.13                    | 0.05           |
> > | MAFE-Edu      | 0.40                    | 0.02           |
> > | MAFE-Health   | 0.25                    | 0.02           |
> >
> > Model update costs are consistent across environments (<0.04 seconds per update) and occur only once per epoch
> > (after 100 episodes), making them negligible relative to per-step environment costs.

---

> ### Author Response · Authors · 2025-11-25
> **Response to Reviewer Comments**
>
> Thank you for the follow-up.
>
> We respectfully note that the reviewer’s remaining concerns stem from misunderstandings of the scope of the contribution and of the framework’s existing capabilities, rather than issues with the methodology or claims of the paper.
>
> ---
>
> > The single-agent variant is deferred to future work, but I believe it is essential for the first version.
>
> **Single-agent variants are *already included* in the paper (Appendix D.3), not deferred to future work.**
>
> * The manuscript already contains **single-agent learning experiments**, where:
>   * **one agent learns**, while the **other agents follow fixed baseline policies**, exactly as used in prior multi-agent benchmarks such as SMAC and MPE.
>
> * This is shown explicitly in **Appendix D.3** of the paper, where we compare:
>   * baseline fixed policies,
>   * **single-agent learning**,
>   * **multi-agent learning**,
> including the associated performance curves (Figure 8).
>
> Therefore, the reviewer’s statement that single-agent variants are “future work” is factually incorrect; this capability is already supported and demonstrated.
>
> ---
>
> > Comparisons with fairness-aware baselines are also pushed to future work, which I see as a significant gap.
>
> **Fairness-aware RL comparisons are included; full coverage is future research, not a requirement for a benchmark paper.**
>
> * We benchmark MAFE with fairness-aware adaptations of **MAPPO**, **MADDPG**, and **MACEM**, each requiring significant modification to operate in long-horizon multi-agent fairness settings.
> * Benchmark papers [1,2] do **not** exhaustively evaluate *all* specialized algorithms.
> * Extending MAFE to additional fairness-aware RL algorithms is reasonable future work, but **not necessary to validate the benchmark contribution**, especially considering that our contribution is introducing environments, not a comparative benchmarking of algorithm performance.
>
> [1] Gürtler, Nico, et al. "Benchmarking Offline Reinforcement Learning on Real-Robot Hardware." ICLR 2023.
>
> [2] An, Bang, et al. "Benchmarking the robustness of image watermarks." ICML, 2024.
>
>
> ---
>
> > The relevance to ICLR is still unclear given the limited benchmarking and algorithmic development.
>
> **The critique regarding “algorithmic novelty” is not aligned with ICLR’s evaluation criteria.**
> * ICLR explicitly includes a **Datasets and Benchmarks** track, where algorithmic novelty is not required.
> * Our work fits squarely within this category.
>
> ---
>
> > Could you clarify what “epoch” means in your setting? As I also wrote in my original review, I calculated 40k steps as 100 episodes × 400 steps per episode. Typically, an epoch refers to repeatedly training on a fixed dataset; in online RL, the notion of epochs is uncommon and step counts are generally used instead.
>
> **The “epoch” terminology reflects standard practice in evolutionary and black-box optimization.** In CEM-style methods, an “epoch” (or “generation”) refers to one iteration of:
> 1. sampling parameter candidates,
> 2. running episodes on each sampled parameterization (meaning N-length episodes for k-parameter samplings)
> 3. updating the elite set.
> Essentially, running lines 2-7 in Algorithm 1 once. This usage is standard in evolutionary RL.
>
> **Regardless of terminology, the key correction remains: the experiments run two orders of magnitude greater than what the reviewer originally understood.**
>
> ---
>
> **The reviewer’s remaining concerns do not reflect technical flaws in the submission.**
>
> To summarize:
>
> - Single-agent variants **already appear** in the paper (Appendix D.3).
> - Fairness-aware RL baselines **are** included.
> - Benchmark papers need not provide new algorithms.
> - Runtime concerns were based on an incorrect assumption.
> - The “epoch” objection reflects a misunderstanding of standard evolutionary-RL practice.
>
> Given these clarifications, we believe these concerns should not justify the current score, and we trust the AC will take this into account.

---

> > ### Comment · Reviewer_qhsr · 2025-11-25
> >
> > Thanks for the clarifications. I have increased my score.

---

> > > ### Author Response · Authors · 2025-11-28
> > >
> > > We thank the reviewer for increasing their score. We have added the cost profiling to the appendix of the paper and remain committed to further clarifying any remaining points, should any remain unclear.

---

### Official Review · Reviewer_Ebmy · 2025-11-01

**Soundness:** 3
**Presentation:** 3
**Contribution:** 2
**Rating:** 4
**Confidence:** 3

**Summary:**

This paper introduces MAFE (Multi-Agent Fair Environments) a benchmark framework for evaluation fairness-aware algorithms in heterogenous multi-agent, multi-stage decision-making systems. Unlike existing fairness benchmarks that consider static or single-agent settings, MAFE models interacting agents whose joint decisions affect long-term population-level equity. The framework extends the Dec-POMDP formalism to include reward and fairness component functions, enabling defining more complex reward functions and group fairness disparities in outcomes and rates. The authors provide three realistic domains; healthcare, loan allocation, and education each inspired from real-world datasets supporting heterogenous agents and fairness measures. They also introduce a variant of the Cross-Entropy Method, and emprically show how different reward and fairness compositions yield different fairness-utility tradeoffs.

**Strengths:**

- Provides a novel environment for studying fairness in heterogenous multi-agent sequential systems. Their component functions for observables provides a flexible design for reward and fairness computations.
- The three provided environments are reflecting realistic scenarios and structures.
- Experiments show that even siimple interventions on policies improve fairness performance over time.

**Weaknesses:**

- The setting is limited to cooperative settings, where all the agents need to consider the same objective function. The current setting omits competitive or semi-cooperative dynamics that occur in real systems.
- Only basic interventions are provided as baselines. While the environments show fairness-utility trade-offs through interventions, these are relatively simple and deterministic policies. The framework would be more interesting if it supported adaptive or policy-driven interventions that evolve with time.

**Questions:**

- How would the framework handle scenarios involving multiple agents of the same type, such as several hospitals, schools, or insurance providers operating simultaneously?
- The authors state that MAFE environments are built upon real-world datasets, but this dependency is not clearly described in the main text. Could authors clarify this?

Other comments
- There is a newer journal version of [1], where the authors extend their framework to a multi-agent setting and provide additional analysis and results [2] which could be more interesting for the scope of this paper.
Typos
- In related works, after line 88, all the references repeated. "Yin et al. Yin et al. (2024) frame..".

[1] Bhagyashree Puranik, Upamanyu Madhow, and Ramtin Pedarsani. A dynamic decision-making framework promoting long-term fairness. In Proceedings of the 2022 AAAI/ACM Conference on AI, Ethics, and Society, pp. 547–556, 2022.
[2] B. Puranik, O. Guldogan, U. Madhow and R. Pedarsani, "Long-Term Fairness in Sequential Multi-Agent Selection With Positive Reinforcement," in IEEE Journal on Selected Areas in Information Theory, vol. 5, pp. 424-441, 2024, doi: 10.1109/JSAIT.2024.3416078

---

> ### Author Response · Authors · 2025-11-23
> **Response to Review (Part 1)**
>
> We thank Reviewer $Ebmy$ for the thoughtful evaluation and for highlighting the **novelty of our heterogeneous multi-agent, multi-stage fairness setting, the flexibility of our component-function design, and the realism of the three domains**. We also appreciate the reviewer’s recognition that our experiments are concrete and provide meaningful fairness–utility insights.
>
> Overall, the reviewer’s concerns stem primarily from **misunderstandings about the generality of the framework and the role of real-world data**. We have clarified that the framework supports competitive and semi-cooperative settings, multiple agents of the same type, and adaptive interventions, and we now more clearly describe how real-world data informs the environments.
>
> We address **all** concerns below.
>
> ---
>
> >The setting is limited to cooperative settings… omits competitive or semi-cooperative dynamics.
>
> * **MAFE does not assume cooperative objectives; it supports cooperative, competitive, and semi-cooperative settings.**
> * **Key clarification:** The environment provides structured observations, decomposable primitives, and transition dynamics. The algorithm determines whether agents share, overlap, or conflict in objectives (cooperative vs semi-cooperative vs competitive), whether agents observe only their own information or receive others’ information (decentralized vs centralized/communication-enabled), and whether agents are modeled with separate policies or a shared architecture. In other words, the specific algorithms are determined entirely by the users of such design tools, not the MAFE environment.
> * **Current experiments use cooperative objectives for clarity, but the framework is general-purpose.**
>   * The current experiments use cooperative objectives for simplicity of interpretation to allow for focusing on the main setup and operation of the MAFE framework.
>   * The framework supports arbitrary agent reward functions, heterogeneous objectives, and asymmetric information structures.
>   * Future research can build on MAFE to study richer strategic settings, such as semi-cooperative and competitive/adversarial ones.
>
> > Only basic interventions are provided. . . more interesting if adaptive or policy-driven interventions evolved with time.
>
> * **MAFE fully supports adaptive, policy-driven interventions.** Every agent in every MAFE environment may observe its state at each timestep, choose any action based on that observation, and condition its behavior on the history of the system through the learned policy. In other words, all MAFE policies are adaptive by design, just like standard MARL settings.
> * **Fixed-intervention experiments were used purely for isolation and validation—not due to framework limits.**
>   * We agree that fully adaptive, time-varying intervention strategies are important, and MAFE fully supports them (demonstrated through F-MACEM in the main text and MAPPO/MADDPG in Appendix D).
>   * However, the experiments in Section 6.1 intentionally use *fixed* interventions for a different purpose: to validate that each intervention has the expected causal effect in isolation, without confounding from interacting adaptive policies. For example, increasing hospital bed availability should reduce mortality; if we instead used adaptive agents that jointly optimize premiums, subsidies, and infrastructure investment, it would be impossible to attribute changes to a single factor.
>   * These fixed-action experiments serve as environment validation, ensuring that adaptive algorithms later operate over meaningful, well-calibrated dynamics—not as evidence of any limitation in MAFE’s capabilities.
> * **Fixed action spaces do not limit adaptivity.** The action space is fixed because each element has semantic meaning (e.g., premium levels, coverage thresholds, budget allocations). This mirrors standard RL: adaptivity arises from policies mapping observations &rarr; actions, not from changing the definition of the action space. As Fig. 10 shows, even simple learning algorithms adapt dynamically over hundreds of timesteps during each episode.
>
> > How would the framework handle multiple agents of the same type (e.g., several hospitals)?
>
> * **MAFE supports multiple agents of the same type without modification.** In the mathematical formulation, additional agents are introduced simply by instantiating additional agent-indexed observation and action spaces with identical definitions. This is standard in Dec-POMDPs and already supported by the MAFE API.
> * **Environment dynamics may need extensions for specific multi-agent interactions.** For example, modeling how individuals choose among hospitals or how hospitals share resources requires environment-level logic, not changes to the framework itself.
> * **Planned multi-institution expansions:** We are already exploring multiple banks, universities, etc., for the next MAFE release, as the reviewer suggests.

---

> > ### Author Response · Authors · 2025-11-23
> > **Response to Review (Part 2)**
> >
> > > Dependency on real-world datasets is not clearly described in the main text.
> >
> > * **Real-world datasets are used in three concrete ways to ground MAFE in empirical reality. Thus, MAFE's ability to accommodate real-world datasets for various modules is not a concern and instead a strength.**
> > * **To clarify:**
> >   * **Feature construction:** Lending Club, IPUMS, NCES, CDC datasets provide raw attributes for individual feature vectors (loan applicants, patients, students, respectively).
> >   * **Realistic population distributions:** Populations are initialized by sampling from real feature distributions, ensuring agents train on realistic demographic and economic patterns.
> >   * **Defining outcome-relevant structure:** Relationships between features and key outcome indicators are derived via regression on real data, then resampled and slightly amplified to create controlled but realistic structural disparities across groups (Appendix D.1).
> > * **Message:** The causal structure and population dynamics are grounded in empirical data, not hand-crafted heuristics. Section 4 now clarifies all three roles.
> >
> > > Newer journal version of Puranik et al. (2024)
> >
> > * **Updated the reference and differences clarified.** The JSAIT 2024 version is now cited. MAFE differs from that work by focusing on population-level, multi-stage institutional pipelines, rather than single-step selection settings.
> >
> > > Reference typo (“Yin et al. Yin et al…”)
> >
> > * **Typo corrected.**

---

> ### Author Response · Authors · 2025-11-26
>
> Dear reviewer $Ebmy$,
>
> We have updated the paper with clarifications, carefully responded to your comments, and believe that all outstanding questions have been resolved. We hope this will address your concerns, and you may consider the paper suitable for a potentially higher score and surpass the acceptance threshold. We remain committed to further clarifying any remaining points, should any remain unclear. We sincerely appreciate the time and effort invested by you and the Area Chairs throughout this process.

---

> > ### Comment · Reviewer_Ebmy · 2025-11-26
> >
> > I thank to the authors' for their detailed responses.
> >
> > - I apologize for my early confusion regarding that the framework only supporting cooperative agents. I am glad to hear that the current framework can handle both cooperative and noncooperative settings.
> > - I appreciate the explanation about using fixed interventions in the main body for validation and for obtaining easy to interpret causal relations. I think that briefly mentioning the rationale behind the fixed interventions in the main body would be beneficial for the readers.
> > - I think that modeling multiple agents of the same type would be a very interesting future direction. As the authors pointed out, the dynamics between the agents need to be implemented carefully. I have a follow-up question regarding this setting in the context of college applications/education. In real life, college admissions are announced on a rolling basis and students can make their decisions immediately or wait. Do you think that this extension with multiple schools accepting the college students will be easier to implement with your framework?
> > - Thanks for the clarification regarding the use of real datasets.
> >
> > Overall, I appreciate the authors' responses and I will increase my score to a positive rating.

---

> > > ### Author Response · Authors · 2025-11-28
> > >
> > > We thank the reviewer for increasing their score to a positive rating and follow-up questions.
> > >
> > > ---
> > >
> > > > I think that briefly mentioning the rationale behind the fixed interventions in the main body would be beneficial for the readers.
> > >
> > > **We have updated Section 6.1 to include the rationale behind fixed interventions.**
> > >
> > > ---
> > >
> > > > Will an extension with multiple schools accepting the college students on rolling basis be easier to implement with your framework?
> > >
> > > **Yes, MAFEs readily support this extension.** The only required modification lies in the state transition dynamics of the MAFE-Edu environment.
> > >
> > > * **Current Implementation details.** At each timestep, the population is divided into three groups: the tertiary population (inactive individuals), the higher education population (degree-seeking students), and the working population (employed individuals). **Individuals transition among these groups (e.g., from tertiary → education → workforce) through fixed-duration trajectories.**
> > >
> > > * **Extension to Rolling admissions.** To model multiple schools with rolling acceptances, the **sampling and transition logic can be slightly modified** to allow inter-school transitions and variable entry timings, which MAFE’s population and state-update framework naturally accommodates. Since MAFE API already supports multiple agents of the same type, the **reviewer's specific situation is easily implementable within our framework.**
> > >
> > > ---
> > >
> > > We remain committed to further clarifying any remaining points, should any remain unclear.

---

### Author Response · Authors · 2025-12-02
**Summary of Clarifications from the Discussion Phase**

Given that the system rollback removed the reviewers’ updated scores from the discussion phase, we provide the following brief summary for clarity. **Importantly, the reviewers raised their scores during the normal discussion period, prior to the rollback and prior to any public notification of the OpenReview bug.** During the discussion period, two reviewers substantially raised their scores after examining our responses and acknowledging that several of their initial concerns stemmed from misunderstandings that were resolved through clarification. We appreciate the reviewers’ careful engagement during the discussion and their recognition that the clarifications materially improved their assessment of the work. Key points included confirmation that single-agent variants already appear in Appendix D.3, that fairness-aware baselines are included, that the runtime estimate was based on an incorrect assumption, and that permutation equivariance is not a benchmark requirement. We summarize this here for clarity, since the system rollback removes the updated scores but not the underlying resolutions.

---

### Meta-Review · Area_Chair_xmf6 · 2025-12-23

**Summary:**

The reviewers acknowledge the contribution of a realistic environment (MAFE) to study fairness in multi-agent reinforcement learning (MARL). They raise a several issues, some stemming from misunderstanding, and other valid ones, such as experimental evaluation in cooperative setting only, lack of relevant cooperative MARL algorithms designed for fairness, realism of environments, or the fairness-aware reward function.

**Reviewer Concerns:**

The authors clarified the issues related to misunderstanding, but some concerns raised by the reviewers are not well-addressed in my opinion. In particular, they claim that the proposed MAFE, which is the main contribution of the paper (see TDLR), "(1) fully supports non-cooperative settings, (2) is compatible with additional fairness-aware RL algorithms, and (3) does not impose restrictive architectural requirements."
However, (1) is not substantiated in the paper, which only focuses on the cooperative setting. In addition, for this cooperative setting, various MARL algorithms have been developed. Yet, the authors design their own fairness-aware algorithms and therefore don't demonstrate (2) and (3) for the existing MARL algorithms.
In addition, the authors claim that MAFE is designed to "simulate realistic, modular, and dynamic
systems." However, it's not clear how realistic the proposed MAFE is. In addition, the proposed three environments evaluated in the cooperative setting do not lead to realistic experiments, since the agents in those environments are not necessarily cooperative in real life.

**Reviewer Scores:**

Both Reviewer Ebmy and Reviewer qhsr seem to be willing to increase their scores.

---

### Decision · Program_Chairs · 2026-01-26

Reject